# Induction of HIV-1-specific antibody-mediated effector functions by native-like envelope trimers in humans

Marloes Grobben[1,2], Emma I. M. M. Reiss[1,2], Angela I. Schriek[1,2], Karlijn van der Straten[1,2,3], Nathalie Dereuddre-Bosquet[4], Pauline Maisonnasse[4], Alex Rooker[1,2], Chunhao Yu[1,2], Khadija Tejjani[1,2], Monica Tolazzi[5], Kwinten Sliepen[1,2], Réka Felfödiné Lévai[6,7], Attila Farsang[6], Roger Le Grand[4], Gabriella Scarlatti[5], Robin J. Shattock[8], Steven W. de Taeye[1,2], Godelieve J. de Bree[2,3], Rogier W. Sanders[1,2,9‡], Marit J. van Gils[1,2‡]*

1 Department of Medical Microbiology, Amsterdam UMC, University of Amsterdam, Amsterdam, The Netherlands, 2 Amsterdam institute for Infection and Immunity, Infectious Diseases, Amsterdam, The Netherlands, 3 Department of Internal Medicine, Amsterdam UMC, University of Amsterdam, Amsterdam, The Netherlands, 4 Université Paris-Saclay, Inserm, CEA, Center for Immunology of Viral, Auto-Immune, Hematological and Bacterial Diseases (IMVA-HB/IDMIT), Fontenay-aux-Roses, France, 5 Viral Evolution and Transmission Unit, Division of Immunology, Transplantation and Infectious Diseases, IRCCS Ospedale San Raffaele, Milan, Italy, 6 Control Laboratory of Veterinary Medicinal Products and Animal Facility, Directorate of Veterinary Medicinal Products, National Food Chain Safety Office, Budapest, Hungary, 7 Ceva-Phylaxia R&D Center, Viral Seed and Antigen Laboratory, Budapest, Hungary, 8 Division of Mucosal Infection and Immunity, Department of Medicine, Imperial College of Science, Technology and Medicine, London, United Kingdom, 9 Department of Microbiology and Immunology, Weill Medical College of Cornell University, New York, New York, United States of America

‡ RWS and MJvG contributed equally to this work as senior authors.
* m.j.vangils@amsterdamumc.nl

## Abstract

A protective vaccine is urgently needed to curb the ongoing global HIV-1 epidemic. There is increased interest to develop a vaccine able to induce both neutralizing antibodies and antibody-mediated effector functions for additional efficacy. We investigated the ability of a group M consensus envelope glycoprotein (Env) trimer vaccine ConM SOSIP.v7 to induce antibodies that mediate effector functions in preclinical and clinical studies. We found that the ConM SOSIP.v7 protein immunogen in combination with MPLA adjuvant induced diverse antibody-mediated effector functions in human volunteers participating in a phase 1 trial. Moreover, the functional antibody response was higher in female compared to male participants. The same immunogen induced similar antibody-mediated effector functions in preclinical studies using rabbits and non-human primates. In these preclinical models, we demonstrated that alterations in the vaccine regimen, including immunization route and adjuvant, could modulate vaccine immunogenicity and lead to functionally different antibody responses. Specifically, we observed that intramuscular immunization led to more functional antibody responses compared to subcutaneous vaccine administration, and that the MPLA liposomes and squalene emulsion adjuvants induced functionally

**Data availability statement:** All relevant data are within the manuscript and its Supporting information files.

**Funding:** This research was funded by an AMC Fellowship (to M.J.v.G.) and an NWO Aspasia grant (to M.J.v.G.). S.W.d.T. is supported by Young investigator grant P-53301 Aidsfonds. The non-human primate study was supported by the European Union's Horizon 2020 (EAVI2020, grant N°681137), the European Infrastructure TRANSVAC2 (grant N°730964), the Agence Nationale de Recherche sur le SIDA et les hépatites virales (ANRS) and the Bill and Melinda Gates Foundation (OPP1115782). G.S., R.W.S, and M.J.v.G. received charitable donations from Fondation Dormeur, Vaduz, for instruments supporting the research of this study. The Infectious Disease Models and Innovative Therapies Research Infrastructure (IDMIT) is supported by the ''Programmes Investissements d'Avenir'' (PIA), managed by the ANR under references ANR-11-INBS-0008 and ANR-10-EQPX-02-01. The funders had no role in study design, data collection and analysis, decision to publish, or preparation of the manuscript.

**Competing interests:** The authors have declared that no competing interests exist.

different antibody responses. In conclusion, this study shows that HIV-1 native-like Env trimers are able to elicit antibody-mediated effector functions in humans and that preclinical studies had predictive value. Furthermore, the preclinical studies revealed that different vaccine formulations and administration routes yield qualitatively different antibody-mediated effector functions. Our findings should guide interpretation of preclinical HIV-1 vaccine studies and can inform the design of HIV-1 vaccine regimens aimed at inducing antibody-mediated effector functions in addition to neutralization capacity.

## Author summary

HIV-1 vaccine development is largely focused on the induction of broadly neutralizing antibodies, with an important role being reserved for stabilized, native-like envelope glycoprotein trimers. The elicitation of antibody-mediated effector functions in combination with broadly neutralizing capacity could provide a benefit towards an efficacious vaccine. We demonstrate that native-like envelope glycoprotein trimers induced several closely coordinated antibody-mediated effector functions in a phase 1 clinical trial in human volunteers. Remarkably, females developed substantially greater functional responses compared to male participants. We found largely similar functional responses in matched preclinical studies in rabbits and non-human primates, providing information that can be taken into account when translating preclinical findings to the clinic. Finally, we also showed that different administration methods and adjuvants modulate functional antibody responses. An eventual HIV-1 vaccine will likely require coordination of all aspects of the immune response. These findings inform the design of vaccination regimens that maximize the induction of diverse antibody-mediated effector functions in supplement to neutralization capacity.

## Introduction

The ongoing global HIV-1 epidemic has caused over 40 million deaths. Despite the availability of antiretroviral therapy (ART) to prevent the spread of HIV-1, 1.3 million new infections occurred in 2022 (https://www.unaids.org/). To overturn the HIV-1 epidemic, the development of a protective vaccine for HIV-1 is essential. The recent Antibody Mediated Protection (AMP) studies demonstrated that neutralizing antibodies (NAbs) against the trimeric envelope glycoprotein (Env) of HIV-1 can protect from infection [1], although NAbs need to be present at high titer. The elicitation of potent and broad NAbs (bNAbs) is therefore a major goal in vaccine development. In recent years, much progress has been made on the generation of stabilized Env immunogens that are capable of eliciting NAbs. These native-like Env trimers, such as the SOSIP immunogens, are now under investigation in several phase 1 clinical trials [2].

While neutralization is expected to be an important correlate of protection for an eventual HIV-1 vaccine [3], antibodies can also mediate effector functions and these can contribute to the protective capacity of bNAbs [4,5]. The antibody Fc tail is able to facilitate a variety of immune cell-mediated functions that can aid in clearance of virus particles as well as HIV-1 infected cells. The composition of the antibody's Fc tail, including the level and type of glycosylation, allelic variations and antibody isotype and subtype, is an important determinant for functionality. The most notable functions include antibody-dependent cellular phagocytosis (ADCP), antibody dependent cellular cytotoxicity (ADCC) and complement dependent cytotoxicity (CDC). The potency and type of effector functions that are induced are determined by interactions with a variety of activating (FcγRI, FcγRIIa, FcγRIIIa, FcγRIIIb) and inhibiting (FcγRIIb) Fc gamma receptors (FcγRs) on different effector cell types, as well as binding to other immune proteins such as the complement protein C1q. Antibody-mediated effector functions can play a role in protection by vaccines against Lassa virus, Malaria and Influenza virus, among others [6–8]. NHP studies also identified antibody effector functions as a correlate of protection following HIV-1 vaccination [9–11]. Moreover, several Fc functions and features were previously shown to be associated with decreased HIV-1 risk, albeit in clinical trials with either limited efficacy or lack of overall efficacy [12,13]. Additional information on the ability of vaccine-elicited antibodies to mediate effector functions can be valuable for the design of successful vaccination strategies.

A major hurdle in the development of an HIV vaccine is the immense genetic variety of >30% in *env [14]*. A possible strategy to overcome this variety is the use of consensus sequences for vaccine design. These synthetic sequences, composed of the most frequently found amino acids at every position, might favor cross-reactive responses to the most conserved sites [15,16]. About 90% of HIV-1 infections are caused by the major (M) group, which has nine subtypes. ConM is the consensus sequence of all consensus sequences from the different clades in the M group. The ConM sequence, when made as recombinant protein using stabilizing mutations, yields native-like Env trimers efficiently [17]. Moreover, the ConM SOSIP.v7 trimer immunogen induced strong autologous NAb responses in rabbits and non-human primates (NHPs) [18–21]. This ConM-based protein immunogen is now under investigation in several clinical trials (ClinicalTrials. gov Identifiers NCT03961438, NCT03816137, NCT04046978 and NCT05208125). We previously performed a phase I clinical trial to study safety and immunogenicity of the ConM SOSIP.v7 Env trimer in HIV-1 uninfected adults and demonstrated that it induced a potent autologous NAb response [22]. In this clinical trial, female participants developed higher neutralization titers and also distinct IgG1 and IgG4 antibody subclass profiles compared to male participants, leading us to hypothesize that functional differences between sexes after ConM SOSIP.v7 protein vaccination may exist. Therefore, we perceived that further investigation of different antibody parameters elicited by the ConM SOSIP.v7 protein immunogen was warranted.

Animal models are extremely important for the assessment and selection of vaccination strategies, including the selection of different adjuvants, administration routes and immunogen formats. However, we know surprisingly little about the translatability of results from preclinical models to humans regarding different antibody functionalities. NHPs such as rhesus or cynomolgus macaques are often the animal model of choice for preclinical vaccine studies andcomprise the last testing phase before entering clinical studies. The NHP immune system is largely comparable to that of humans and they have similar FcγRs to humans [23]. NHPs also have four IgG subclasses, but they are structurally and functionally more similar to each other than the human versions [23]. Smaller animals allow more extensive and cost-effective testing in early-stage preclinical studies. In the HIV-1 field, rabbits are generally favored instead of mice because they have a more similar distribution of antibody complementarity-determining region lengths compared to humans, allowing the induction of NAbs [24]. Very little is known regarding antibody-mediated effector functions in rabbits, however rabbit serum antibodies are able to bind human FcγRs and induce effector functions in *in vitro* assays using human cells [25–27]. As the ConM SOSIP.v7 Env trimer immunogen passed preclinical testing in rabbit and macaques and testing in humans, we had the unique opportunity to study whether antibody-mediated effector functions observed in rabbits and macaques can be predictive for results in humans.

Additional knowledge on how to optimally induce these antibody-mediated effector functions is also needed. Adjuvants are known to augment and steer T cell responses and also strengthen and modulate NAb responses, including NAb breadth and longevity [28]. Traditional adjuvants such as alum or oil-in-water emulsions have such effects, but newer adjuvants including those activating toll-like receptors (TLRs), are more powerful and tunable [29]. The administration route of a vaccine is also a major contributing factor in vaccine responses, but little is known about the effect on the generation of Fc functions [29,30]. More knowledge on the effect of adjuvants and administration route on the induction of antibody-mediated effector functions after HIV-1 vaccination will be beneficial for future human studies.

Here, we investigated the ability of the ConM Env trimer immunogen to induce antibody-mediated effector functions. Using several *in vitro* assays, we established that ConM SOSIP.v7 protein vaccination elicits antibodies that are able to induce effector functions in humans, with substantial differences between female and male participants. Moreover, we assessed how these results compared to those induced in two preclinical studies in rabbits and macaques. In the preclinical studies, we also demonstrated the efficient induction of several antibody-mediated effector functions and we observed that changes in adjuvant or administration route can alter the Fc functionality of the elicited antibodies. Our findings can inform improved HIV-1 vaccine regimens aimed at inducing antibody-mediated effector functions in addition to neutralization capacity.

## Results

### The ConM SOSIP.v7 protein vaccine with MPLA liposomes elicits antibody-mediated effector functions in humans

We characterized the effector function profile of antibodies induced in humans by vaccination with the HIV-1 Env immunogen ConM SOSIP.v7 [17]. In ACTHIVE-001, a randomized, open-labelled, uncontrolled phase 1 clinical trial, 23 HIV-1 seronegative individuals in general good health were vaccinated three times with the stabilized Env protein ConM SOSIP.v7 at weeks 0, 8 and 24 (Fig 1A). The protein vaccine was adjuvanted with monophosphoryl lipid A (MPLA), a TLR4 agonist, formulated with liposomes. Thirteen individuals received three full dose vaccinations of 100 µg ConM SOSIP.v7 while 11 individuals received two full dose vaccinations followed by a fractional dose of 20 µg to see if this fractional dose may enhance NAb induction. All vaccine dosages were adjuvanted with a consistent dose of 500 µg MPLA liposomes. Safety and immunogenicity, including NAb titers, are described by Reiss *et al.*[22]. Females had significantly higher NAb titers compared to males. No statistically significant differences in IgG levels or Nab titers were observed between the two dosage groups after the third vaccination. Therefore, we did not differentiate between the individuals who did or did not receive the fractional dosing schedule for the analyses here. We first assessed the level of serum ConM-specific IgG induced by the vaccine using the ConM SOSIP.v7 protein conjugated to Luminex beads. Vaccine-specific IgG antibodies were detected in 20/23 individuals after two vaccinations (at week 10), and in all individuals after three vaccinations (at week 26) (Fig 1B).

Then, we investigated to what extent the elicited antibodies could interact with FcγRIIa (linked to ADCP), FcγRIIIa (linked to ADCC/NK cell activity) and C1q (linked to complement activity). For all individuals with detectable ConM-specific antibodies, interaction with FcγRIIa and FcγRIIIa was also detectable (20/23 after two vaccinations and 23/23 after three vaccinations) (Fig 1 and 1D). Interaction with complement protein C1q was detected in 21/23 individuals after two vaccinations and in all individuals after three vaccinations (Fig 1E). IgG levels were strongly correlated with FcγRIIa and FcγRIIIa engagement at weeks 10 (r = 0.96 and 0.95, respectively; P < 0.0001) and 26 (r = 0.87 and 0.77, respectively; P < 0.0001) (S1A and S1B Fig). For C1q, IgG levels were moderately correlated (r = 0.66, P < 0.001) at week 10 and strongly correlated at week 26 (r = 0.78, P < 0.0001) (S1C Fig). We also compared these ConM-specific IgG titers with ConM neutralization titers and found a significant correlation at week 10 (r = 0.77, P < 0.0001) and at week 26 (r = 0.71, P < 0.001) (S1D Fig).

Next, we assessed the ability of serum antibodies to initiate phagocytosis of ConM SOSIP.v7-conjugated beads. ADCP activity was observed after two vaccinations in nearly all individuals and this increased after the third vaccination (Fig 1F). Subsequently, we assessed the ability of serum antibodies to activate NK cells using NK cells isolated from healthy donor

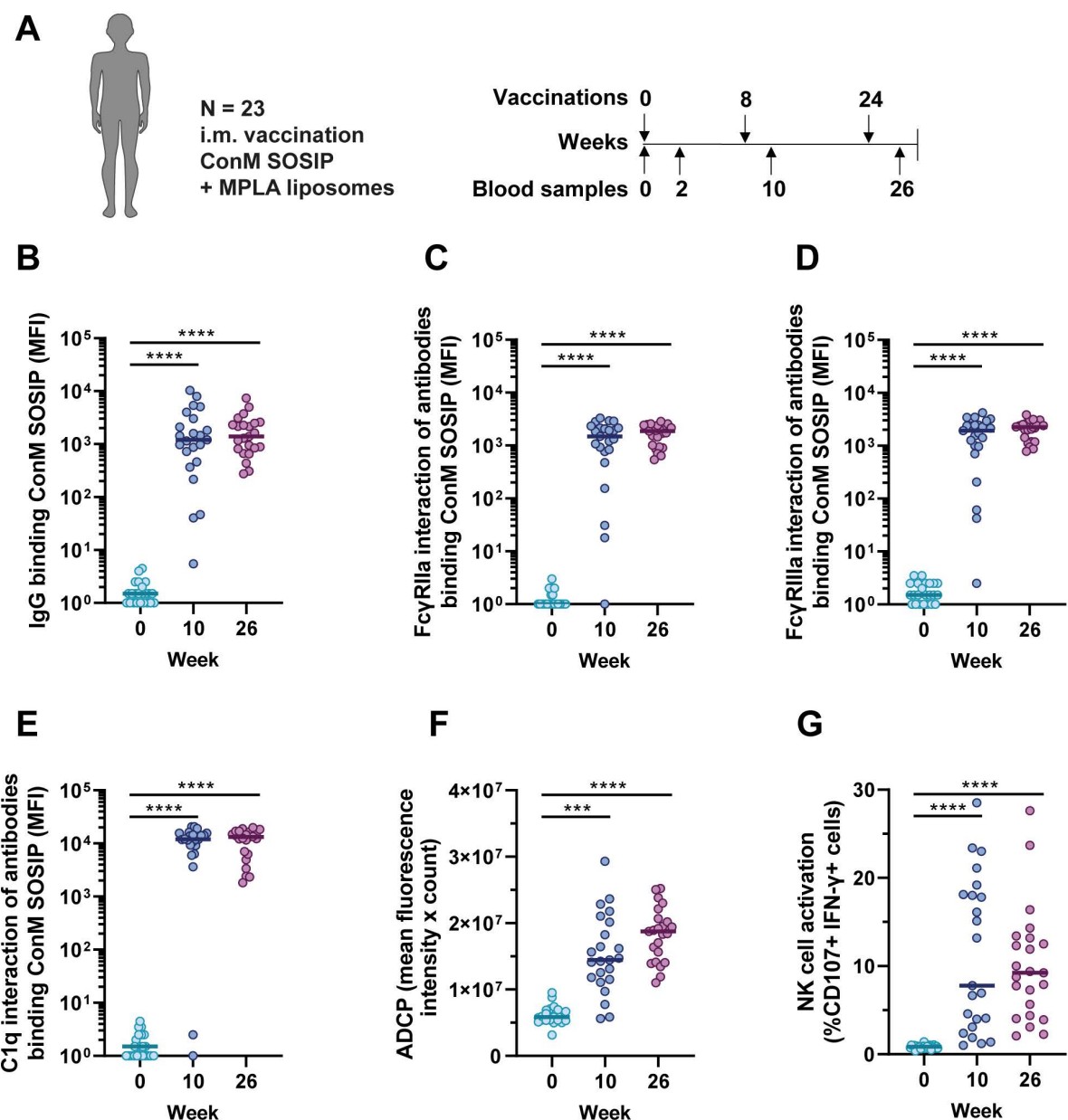

**Fig 1. IgG levels and antibody functionality elicited by the ConM SOSIP.v7 protein vaccine with MPLA liposomes in healthy human volunteers.** (A) Composition of the cohort and the vaccination schedule. Twenty-three individuals received either a full dose regimen or two full doses (100 µg ConM SOSIP.v7) and a fractional third dose (20 µg ConM SOSIP.v7) by the intramuscular (i.m.) route. Individuals from the two groups were pooled since we did not identify any differences between the doses in a prior study. Arrows above the vaccination schedule represent the intramuscular vaccinations and arrows below the schedule represent the blood samples analyzed in the current study. (B) ConM-specific IgG levels in serum at baseline (week 0), after two vaccinations (week 10) and after three vaccinations (week 26). (C) Interaction of ConM-specific antibodies with FcγRIIa, (D) with FcγRIIIa and (E) with C1q. Data for panels B-E was measured by Luminex immunoassay is presented as blank-corrected median fluorescence intensity (MFI). (F) Antibody dependent cellular phagocytosis (ADCP) of ConM SOSIP.v7 conjugated beads by THP-1 cells mediated by serum antibodies at weeks 0, 10, and 26. The ADCP score is composed of the mean fluorescence intensity multiplied by the count of internalized beads. (G) NK cell activation by ConM-specific serum antibodies, presented as the percentage of cells expressing CD107 and IFN-γ. In panels B-G, all three groups were compared with each other using a Friedman test followed by a Dunn's multiple comparison test. The colored horizontal lines represent the medians in each group. *** = $P < 0.001$, **** = $P < 0.0001$.

peripheral blood mononuclear cells (PBMC) and ConM SOSIP.v7 conjugated plates. NK cell activation is a surrogate for ADCC [31]. Sera from most participants were capable of inducing specific upregulation of NK cell activation markers CD107 and IFN-γ after two or three vaccinations, with up to 39% of cells becoming activated (Fig 1G). ADCP and NK cell activation are initiated through antibody interaction with FcγRIIa and FcγRIIIa, respectively. ConM-specific serum antibody interaction with FcγRIIa correlated strongly with ADCP activity (r=0.76, P<0.0001) (S1E Fig) and interaction with FcγRIIIa correlated with NK cell activation (r=0.80, P<0.0001) (S1F Fig). Moreover, both ADCP activity and NK cell activation correlated with IgG1 and IgG3, but not with the IgG2 and IgG4 antibody levels reported by Reiss et al.[22] (S1G and S1H Fig). Taken together, ConM SOSIP.v7 vaccination in healthy adult volunteers resulted in an antibody response with multiple antibody functionalities which appear to be highly coordinated.

## Females induce stronger antibody-mediated effector functions than males in response to the ConM SOSIP.v7 protein vaccine with MPLA liposomes

Female participants had higher ConM-specific NAb responses, IgG and IgG1, while males had stronger ConM-specific IgG4 responses [22]. In the assay used in the current study, median IgG levels were 2.0-fold higher in females (P<0.05) after two vaccinations (week 10) and 2.4-fold higher in females (not significant, P=0.08) after the third vaccination (week 26) (Figs 2B and S2A). At week 26, female participants showed significantly more interaction of ConM-specific serum antibodies with FcγRIIa, FcγRIIIa and C1q (median 1.8-fold, 1.5-fold and 1.6-fold higher in females, respectively) (Fig 2C–E). These differences were also observed in the functional assays with 1.3-fold higher ADCP activity and 2.0-fold higher NK cell activation in females (Fig 2F and 2G). This is consistent with the differences observed in antigen-specific IgG levels and NAb titers previously described [22]. At week 10 we observed similar differences, but with an even larger difference in NK cell activation (5.0-fold higher in females) (S2B–S2G Fig).

To further investigate these differences, we performed a principal component analysis (PCA) to see which of the measured characteristics of the antibody response mostly differed between the two sexes. The PCA of serum responses after three vaccinations showed a separation between females and males on principal component (PC) 1, with a high degree of explained variance (78%), but no clear separation on PC 2 (S3A Fig). The variable loading of PC 1 indicates that the interaction with FcγRIIa, FcγRIIIa and C1q are the strongest contributors, followed by IgG levels and then ADCP. Neutralization and NK cell activation are weaker contributors to the separation on PC 1 and thus differ the least between females and males in this model (S3B Fig). We also performed a PCA of serum responses after two vaccinations. Outcomes were similar but indicate a smaller role for antibodies interacting with C1q and a larger role for IgG levels (S3C and S3D Fig). Together, these results indicate a difference in antibody Fc functionality between sexes, which is in line with the increased occurrence of more functional antibody subclasses observed by Reiss et al.[22].

## The ConM SOSIP.v7 protein vaccine with MPLA liposomes elicits antibody-mediated effector functions in rabbits and macaques

Previously performed preclinical studies, which preceded the clinical trial, provided a unique opportunity to investigate if antibody-mediated effector functions in these animal models could be predictive of the response in humans. Sex was not considered in the study designs since the outcomes of the clinical trial were not yet known. Therefore, only female animals were included. Six female cynomolgus macaques were vaccinated with the same immunogen (ConM SOSIP.v7), formulated with the same adjuvant (MPLA liposomes). The vaccination schedule was identical to the schedule for the full dose group of the human study, with vaccines administered at week 0, week 8 and week 24 (Fig 3A). Likewise, we studied six female rabbits vaccinated with ConM SOSIP.v7 formulated with MPLA liposomes. The rabbit study used a slightly more condensed vaccination schedule, with vaccines administered at week 0, week 4 and week 20. We analyzed blood collected prior to vaccination and 2 weeks following each vaccination. We compared the outcomes in these animal models with the 7 female clinical trial participants receiving the full dose vaccination schedule.

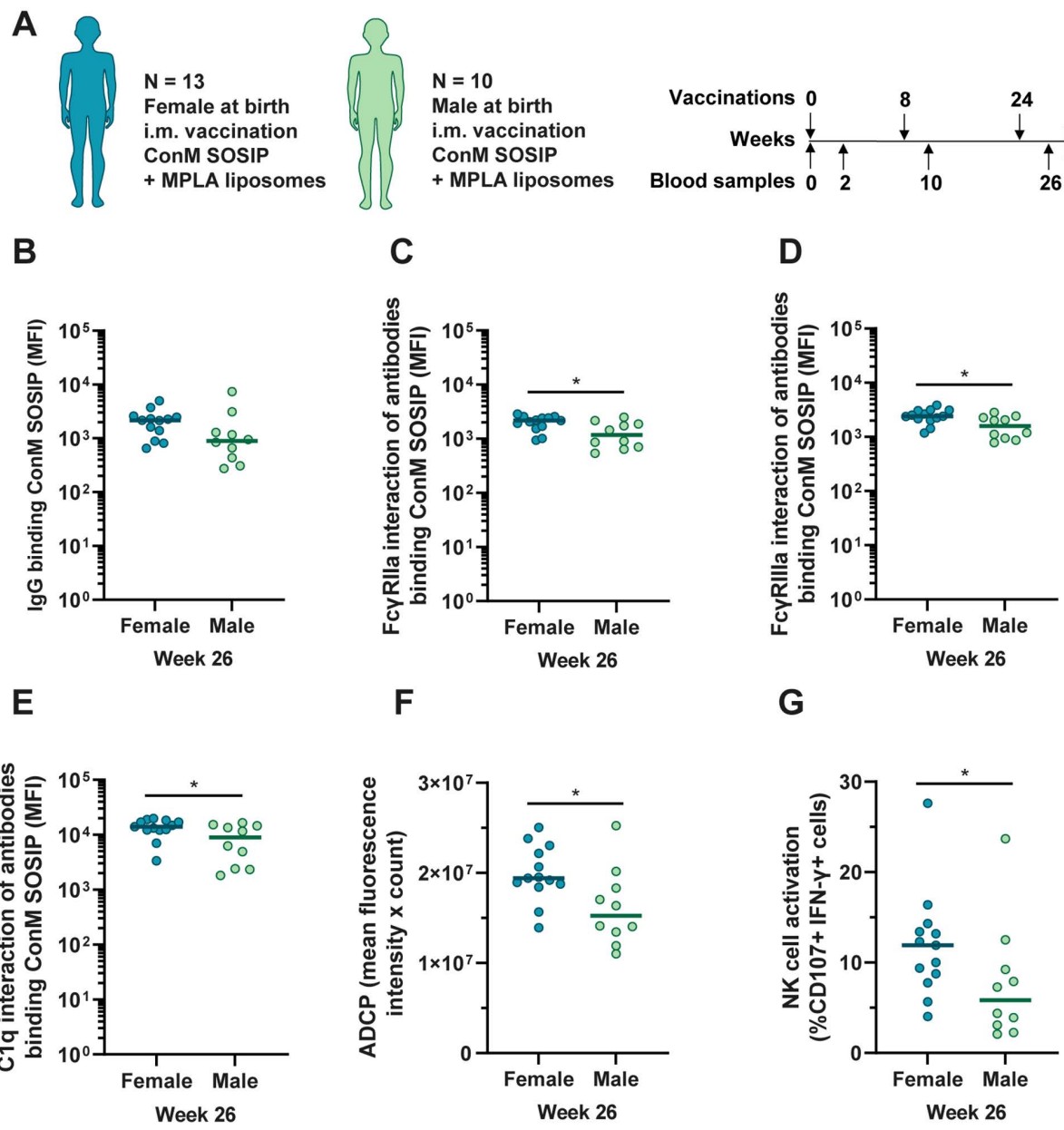

**Fig 2. Comparisons of IgG levels and antibody functionality between clinical trial participants with female and male sex at birth.** (A) Composition of the cohort and vaccination schedule. The cohort included 13 participants with female sex at birth and 10 participants with male sex at birth. Arrows above the vaccination schedule represent the intramuscular vaccinations and arrows below the schedule represent the blood samples analyzed in the current study. (B) ConM-specific IgG levels after three vaccinations (week 26) were compared between female (N = 13) and male (N = 10) ACTHIVE-001 clinical trial participants. (C) Interaction of ConM-specific antibodies with FcγRIIa, (D) with FcγRIIIa and (E) with C1q. Data for panels B-E was measured by Luminex immunoassay is presented as blank-corrected median fluorescence intensity (MFI). (F) Antibody dependent cellular phagocytosis (ADCP) of ConM SOSIP.v7 conjugated beads by THP-1 cells mediated by serum antibodies was compared between female and male participants. The ADCP score is composed of the mean fluorescence intensity multiplied by the count of internalized beads. (G) NK cell activation by ConM-specific serum antibodies, presented as the percentage of cells expressing CD107 and IFN-γ. Baseline and week 10 responses corresponding to panels F and G are shown in S2 Fig. ADCP and NK cell activation normalized by IgG titer are shown in S3 Fig. In panels B-G, groups were compared using a Mann-Whitney U test. The colored horizontal lines represent the medians in each group. * = P < 0.05.

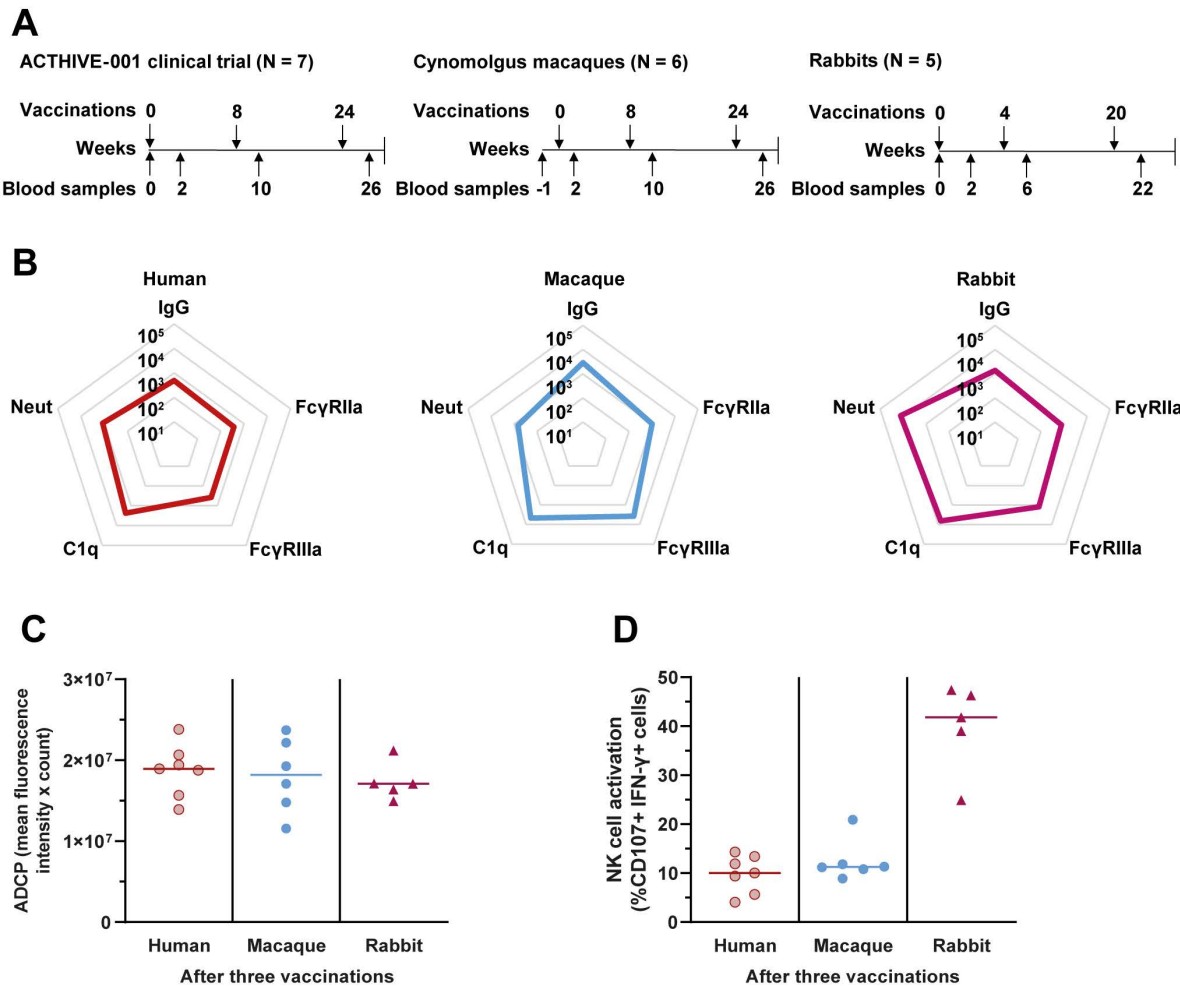

**Fig 3. ConM-specific IgG levels, neutralization and Fc functionality compared between three different species.** (A) Vaccination schedule for each study. All individuals and animals received three full-dose ConM SOSIP.v7 vaccines adjuvanted with MPLA liposomes administered intramuscularly. The vaccination scheme for the macaques and the clinical trial was identical and the rabbits received a slightly condensed vaccination schedule. Only female clinical trial participants who received the full-dose regimen were included in this comparison. Six rabbits were originally included but one died before the week 22 serum could be collected. (B) ConM-specific IgG levels and interaction of ConM-specific antibodies with FcγRIIa, FcγRIIIa and C1q, all expressed as blank-corrected median fluorescence intensity (MFI) measured by Luminex assay, and half-maximal infective dilution (ID$_{50}$) ConM pseudovirus neutralization titers after three vaccinations. Human FcγRs were used for rabbit and human sera, while macaque FcγRs were used for macaque sera. Human C1q protein was used for all species. All outcomes are log-transformed to facilitate comparison between models. An overlay of the three plots can be found in S4 Fig. (C) Antibody dependent cellular phagocytosis (ADCP) of ConM SOSIP.v7 conjugated beads by THP-1 cells mediated by serum antibodies after three vaccinations. The ADCP score is composed of the mean fluorescence intensity multiplied by the count of internalized beads. (D) Activation of NK cells derived from human healthy donor PBMCs by ConM-specific serum antibodies after three vaccinations. The NK cell activation is presented as the percentage of cells expressing CD107 and IFN-γ. Sera of each species were diluted equally to facilitate these comparisons. Since the data comprised different species, the results were assessed qualitatively and without statistical analysis. The colored horizontal lines represent the medians in each group.

We could detect ConM-specific IgG in all species. Furthermore, serum antibodies from rabbits, macaques and humans after three vaccinations had the ability to interact with human and macaque FcγRIIa, FcγRIIIa and human C1q protein (Figs 3B and S4). To qualitatively compare the capacity of serum antibodies to engage FcγRs and C1q across species, we compared ConM-specific binding signals while taking any differences in IgG titers between the species into consideration. The relationship between IgG titers, FcγRIIa, FcγRIIIa and C1q interaction was quite consistent for all three species,

with the exception that FcγRIIa and C1q interaction in macaques was somewhat weaker than expected considering the IgG levels. Neutralizing antibody levels were relatively high in the rabbit studies, and lower in the macaque studies when compared to the human study. ADCP activity mediated by serum was similar for the three models, while human NK cell activation was similar in humans and macaques and highest in rabbits (Fig 3C and 3D). Overall, the observed effector responses were qualitatively similar between species with quantitative differences being observed in some assays. These results suggest that HIV-1 protein vaccination in rabbits and macaques induces antibody-mediated effector functions that could be predictive of these functions in humans.

## Different adjuvants and immunization routes modulate antibody functionality in ConM-vaccinated macaques

Following our demonstration of the induction of ConM-specific antibodies able to mediate effector functions in cynomolgus macaques, we set out to investigate whether this functionality can be modified by alterations in adjuvant type or administration route. There were two additional groups of six female cynomolgus macaques each, vaccinated with the same immunogen (ConM SOSIP.v7). One group received the immunogen formulated with the same adjuvant (MPLA liposomes) but administered s.c., while the other group received the immunogen formulated with another adjuvant, a squalene emulsion similar to the MF59 adjuvant used in humans, administered i.m. The vaccination and sampling schedule was identical (Fig 4A). All three groups were previously tested for, amongst others, serum neutralization capacity [19,20].

After the full vaccination schedule, the i.m. squalene emulsion adjuvanted group had the highest IgG levels with a 2.3-fold increase compared to MPLA liposomes and a statistically significant 13-fold increase compared to the s.c. MPLA adjuvant group (Fig 4B). IgG levels differed remarkably between animals in the s.c. vaccinated group. The i.m. group had 5.8-fold higher IgG levels compared to the s.c. MPLA adjuvanted group (Fig 4C). Next, we characterized the ability of the elicited antibodies to engage macaque FcγRIIa, macaque FcγRIIIa and human C1q protein (Fig 4C–E). After three vaccinations (week 26), these interactions were measurable in all groups and significantly higher for the squalene emulsion adjuvanted i.m. group compared to the MPLA adjuvanted s.c. group (median 20-fold, 19-fold and 253-fold increase, respectively, P < 0.05 for all). There was high heterogeneity between individual macaques in the s.c. vaccinated group. With the MPLA adjuvant, the i.m. vaccination resulted in higher responses than the s.c. vaccination (median 7.2-fold, 6.7-fold and 16-fold increase, respectively, not statistically significant). Macaques vaccinated i.m. with squalene emulsion had higher levels of interaction with macaque FcγRIIa, macaque FcγRIIIa and human C1q protein than the macaques vaccinated i.m. with MPLA (median 2.9-fold, 2.8-fold and 16-fold increase, respectively), albeit not statistically significant. The observed differences were also present already after two vaccinations (S5A–S5D Fig). IgG levels were strongly correlated with antibodies interacting with macaque FcγRIIa and FcγRIIIa (r = 0.81 and 0.84, both P < 0.0001) and also with binding to human C1q (r = 0.82, P < 0.0001) (S6A–S6C Fig). IgG levels also correlated with ConM neutralization titers, but less strongly (r = 0.61, P < 0.01) (S6D Fig). This is in agreement with relative differences between the groups in ConM neutralization capacity as previously published, although the difference between the i.m. groups was very subtle [20]. The magnitude of FcγRIIa and FcγRIIIa interaction were also investigated for antibodies able to bind heterologous antigens, showing that antibodies that interact with FcγRIIa and FcγRIIIa can have some breadth. As expected, levels of interaction with FcγRIIa and FcγRIIIa were markedly lower for the heterologous antigens, especially for the SOSIP trimer derived from the clade A strain BG505 (S7C–S7F Fig).

Next, we investigated if the antibody functionality indicated by FcγR and C1q binding was also detectable using human effector cell assays. After the third vaccination, we detected ADCP activity mediated by human THP-1 cells in all vaccinated macaques, but there was significantly more ADCP activity in the i.m. vaccinated animals compared to the s.c. vaccinated animals (median 1.7-fold increase for both comparisons). There was no difference between the two adjuvants (Fig 4F). ADCP activity correlated with FcγRIIa interaction (r = 0.55, P < 0.05) (S6E Fig). After three vaccinations, macaque sera clearly activated human NK cells. Activation was significantly higher in the squalene emulsion adjuvanted i.m. group compared to the MPLA adjuvanted s.c. group (median 3.5-fold increase, P < 0.05), but not significantly higher when MPLA

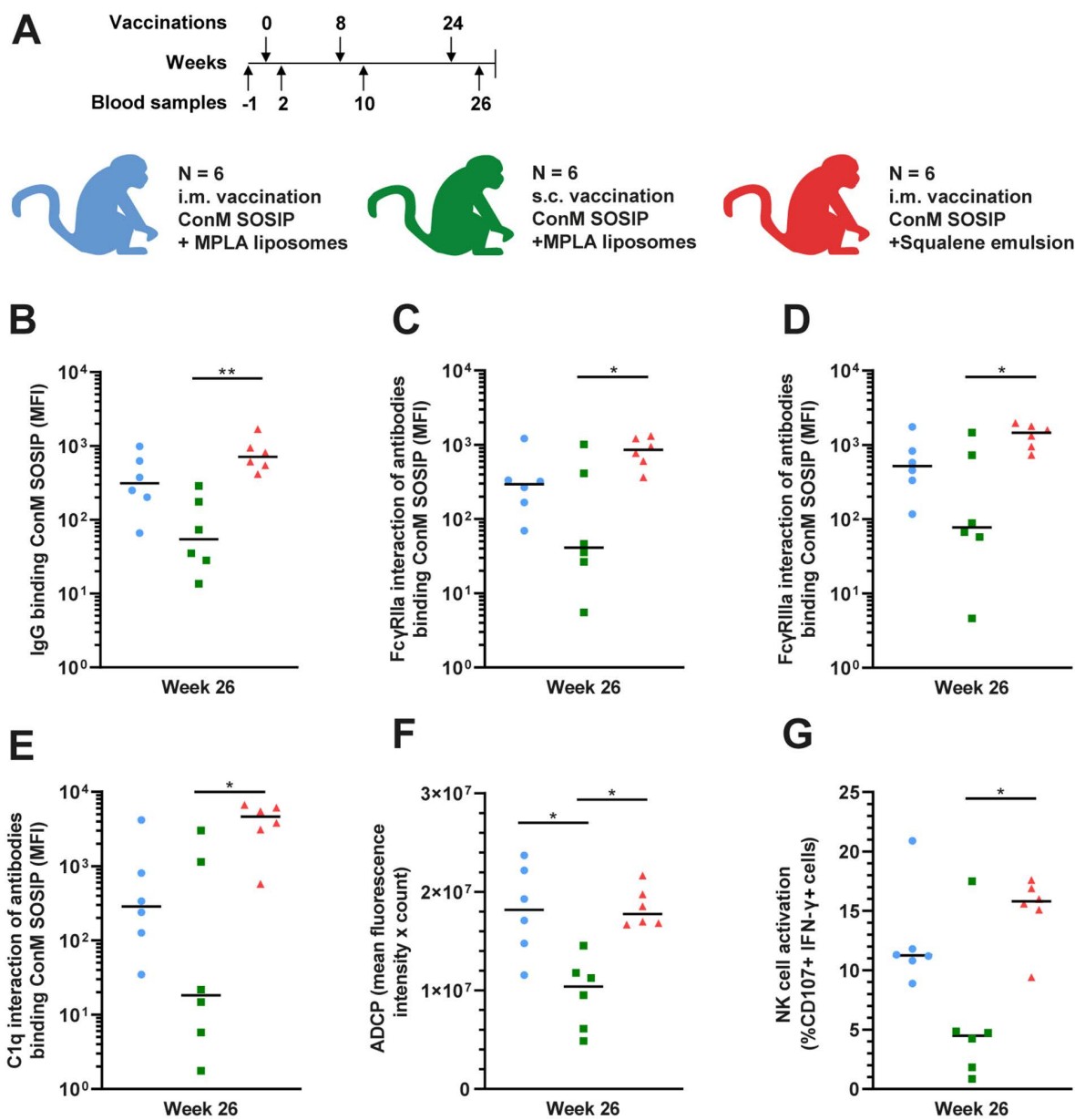

**Fig 4. Comparison of IgG levels and antibody functionality in macaques vaccinated with the ConM SOSIP.v7 protein vaccine using different administration routes and adjuvants.** (A) Composition of the three vaccination groups and the vaccination schedule. Six animals per group, vaccines were administered intramuscularly (i.m.) or subcutaneously (s.c.) and adjuvanted with either MPLA liposomes or squalene emulsion. Arrows above the vaccination schedule represent the vaccinations and arrows below the schedule represent the blood samples analyzed in the current study. (B) ConM-specific IgG levels in serum after three vaccinations (week 26). (C) Interaction of ConM-specific antibodies with cynomolgus macaque FcγRIIa, (D) cynomolgus macaque FcγRIIIa and (E) human C1q. Data for panels B-E was measured by Luminex immunoassay and is presented as blank-corrected median fluorescence intensity (MFI). (F) Antibody dependent cellular phagocytosis (ADCP) of ConM SOSIP.v7 conjugated beads by THP-1 cells mediated by serum antibodies after three vaccinations (week 26). The ADCP score is composed of the mean fluorescence intensity multiplied by the count of internalized beads. (G) NK cell activation by ConM-specific serum antibodies, presented as the percentage of cells expressing CD107 and IFN-γ. Baseline responses corresponding to panels F and G are shown in S5 Fig. ADCP and NK cell activation normalized by IgG titer are shown in S7 Fig. In panels B-G, all three groups were compared with each other using a Kruskal-Wallis test followed by a Dunn's multiple comparison test. The colored horizontal lines represent the medians in each group. *=P<0.05, **=P<0.01.

adjuvanted i.m. vaccination was compared with s.c. vaccination (median 2.5-fold increase) and also not significantly higher when the squalene emulsion adjuvant was compared with MPLA using i.m. vaccination (median 1.4-fold increase) (Fig 4G). NK cell activation correlated strongly with FcγRIIIa interaction (r = 0.86, P < 0.0001) (S6F Fig).

Subsequently, we performed a PCA to define which of the measured antibody-related immune parameters differed the most between the experimental groups (S6G Fig). We observed a separation on PC 1 between the s.c. vaccine group and the two i.m. groups, while the distinction between the MPLA and squalene emulsion-adjuvanted vaccine was more subtle. The results indicated that antibody levels, interactions related to effector functions and NK cell activation all contributed to the separation between the s.c. and i.m. vaccination groups, while differences in neutralization and ADCP were less pronounced (S6H Fig). ADCP was the main contributor to PC2, however there was limited separation between the groups on PC2. Overall, the macaque study showed a clear superiority of i.m. vaccination for induction of antibodies with various functionalities and a subtle non-significant increase in functionality with the squalene emulsion adjuvant.

## Adjuvants modulate antibody functionality in ConM-vaccinated rabbits

To confirm our findings that adjuvants can impact the Fc functionality of antibody responses, we also compared MPLA adjuvanted, squalene emulsion adjuvanted and non-adjuvanted ConM SOSIP.v7 immunization in rabbits. Vaccines were administered at weeks 0, 4 and 20 (Fig 5A). Sera from all three groups were previously tested for, amongst others, antibody neutralization capacity [21]. IgG levels did not reach our threshold for a detectable response in the no adjuvant control group (see methods). The rabbits that received MPLA and squalene emulsion adjuvants had higher IgG levels than those from the no adjuvant group (median 140- and 133-fold increase) (Fig 5B). We also detected significant interaction of ConM-specific antibodies with human FcγRIIa and FcγRIIIa in the MPLA and squalene emulsion-adjuvanted groups, but very low or undetectable interaction in the no adjuvant group (median 489- and 528-fold increase for FcγRIIa and median 666- and 586-fold increase for FcγRIIIa) (Fig 5C and 5D). The animals that did not receive any adjuvant had no ConM-specific antibodies able to interact with human C1q, whereas all animals receiving MPLA or squalene emulsion did have detectable levels of these type of antibodies (Fig 5E). No differences between the adjuvanted groups were detected after three vaccinations. After two vaccinations, responses appeared higher after squalene emulsion adjuvant was used, but this was only significant for C1q interaction (196-fold increase, P < 0.05) (S8D Fig). The observed differences at week 22 were also visible at earlier time points (S8A–S8D Fig). There was a strong correlation between IgG levels and interaction with FcγRIIa, FcγRIIIa and C1q (r = 0.95, 0.99 and 0.92, respectively, all P < 0.0001) (S9A–S9C Fig). Neutralization titers also correlated strongly with IgG levels (r = 0.82, P < 0.0001) (S9D Fig).

To further confirm the functionality of these ConM-specific serum antibodies, we tested their ability to induce ADCP activity by human THP-1 cells (Figs 5F and S8E). The ADCP activity after vaccination without adjuvant was similar to background activity, whereas we observed a clear increase of ADCP activity after three vaccinations (week 22) with a 2.0-fold higher response in the squalene emulsion adjuvanted group and a 1.8-fold higher ADCP response in the MPLA adjuvanted group compared to the no adjuvant group. ADCP activity correlated with FcγRIIa interaction (r = 0.74, P < 0.0001) (S9E Fig). We also tested the ability of ConM-specific serum antibodies to activate human NK cells. We observed no NK cell activation by serum from the no adjuvant group, but strong activation in the groups receiving adjuvanted vaccines. There was a 45-fold and 47-fold increase in NK cell activation when using MPLA and squalene emulsion as adjuvant, respectively (Figs 5G and S8F). NK cell activation correlated strongly with FcγRIIIa interaction (r = 0.88, P < 0.0001) (S9F Fig). We also compared outcomes between the rabbit study with previously discussed outcomes in an overarching correlations analysis, including results from the human and macaque studies. ADCP scores and percentages of activated NK cells were compared with neutralization titers (S9G and S9H Fig). The results show that rabbit responses slightly deviate from the correlations in the macaque and human studies, particularly regarding NK cell activation.

Finally, we also performed a PCA to define which of the measured antibody-related immune parameters in rabbits differ the most between the experimental groups (S9I Fig). We observed a distinct separation on PC 1 between the adjuvanted

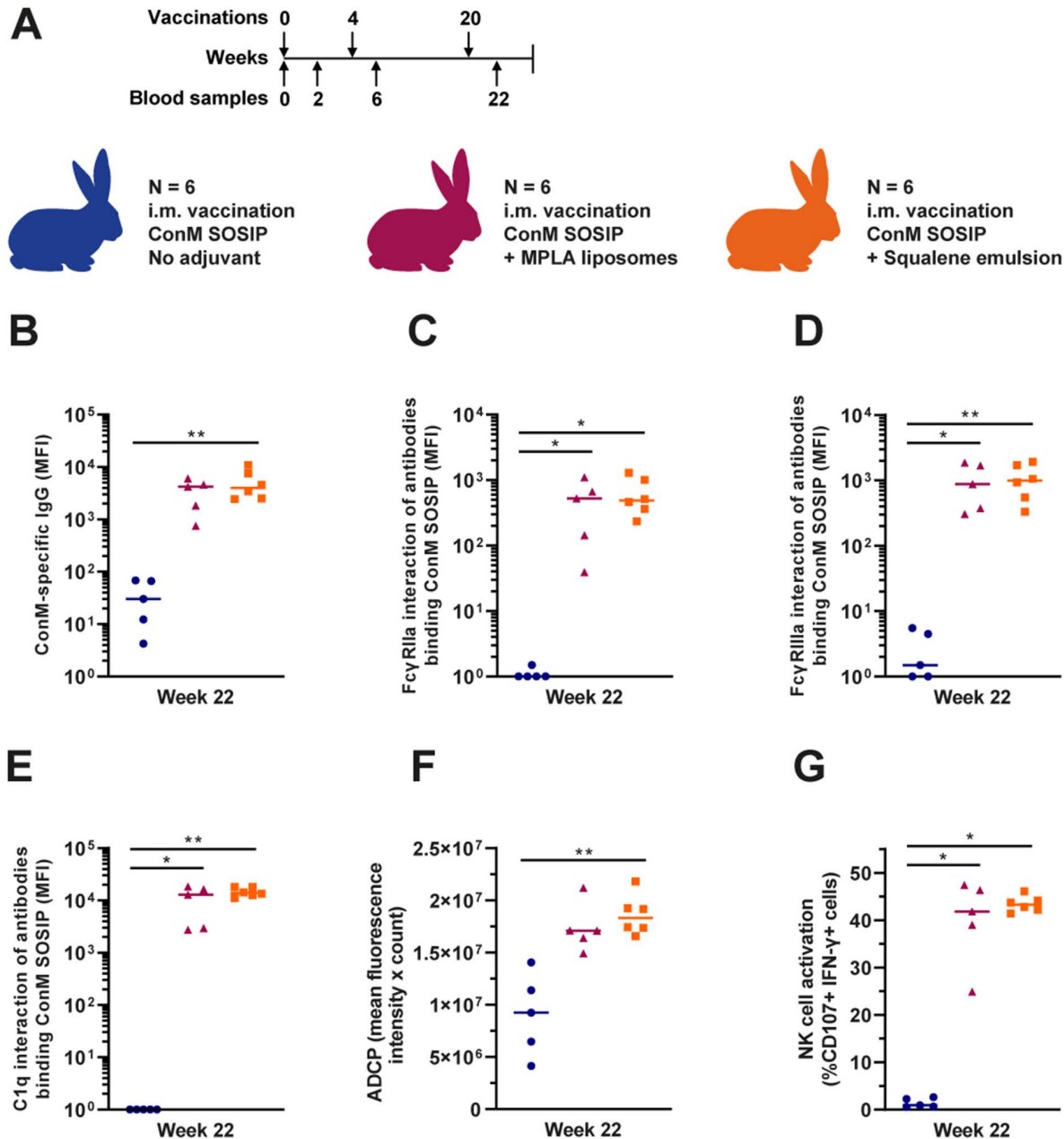

**Fig 5. IgG levels and antibody functionality in rabbits vaccinated with the ConM SOSIP.v7 protein vaccine using different adjuvants.** (A) Composition of the three vaccination groups and the vaccination schedule. Six animals per group, vaccines were administered intramuscularly (i.m.) with either MPLA liposomes, squalene emulsion or no adjuvant. Arrows above the vaccination schedule represent the vaccinations and arrows below the schedule represent the blood samples analyzed in the current study. One rabbit from the no adjuvant control group and one rabbit from the MPLA adjuvant group died before the week 22 sera were collected. (B) ConM-specific IgG levels in serum after three vaccinations (week 22). (C) Interaction of ConM-specific antibodies with human FcγRIIa, (D) human FcγRIIIa and (E) human C1q. Data for panels B-E was measured by Luminex immunoassay and is presented as blank-corrected median fluorescence intensity (MFI). (F) Antibody dependent cellular phagocytosis (ADCP) of ConM SOSIP.v7 conjugated beads by THP-1 cells mediated by serum antibodies after three vaccinations (week 22). The ADCP score is composed of the mean fluorescence intensity multiplied by the count of internalized beads. (G) NK cell activation by ConM-specific serum antibodies, presented as the percentage of cells expressing CD107 and IFN-γ. In panels B-G, three groups were compared with each other using a Kruskal-Wallis test followed by a Dunn's multiple comparison test. The colored horizontal lines represent the medians in each group. *=P<0.05, **=P<0.01.

groups and the no adjuvant group, while there was no distinction between the MPLA and squalene emulsion recipient rabbits. Nearly all measurements contributed equally to the observed separation. This indicates that both antibody levels and interactions related to effector functions significantly differ, but differences in neutralization and ADCP were contributing less to the difference on PC 1 (S9J Fig). Neutralization and ADCP were a major contributor to PC 2, however there was no separation between the groups on PC 2. Overall, the rabbit study showed a clear effect of adjuvants on the induction of antibody effector functions. However, the trend for increased functionality that we observed with the squalene emulsion adjuvant in the macaques was only visible after two vaccinations in the rabbits.

## Discussion

The development of an HIV-1 vaccine has proven to be a formidable yet unsolved challenge, in which likely all aspects of the immune response need to be carefully tuned to increase the potential for efficacy. Therefore, there is increased interest for obtaining a more complete picture of the immune response, including antibody-mediated effector functions. It is important to establish whether these functions are consistently induced by vaccination. Here, we show that vaccination with a HIV-1 native-like trimer vaccine, ConM SOSIP.v7, induces antibodies that can potently engage FcγRs and C1q and induce antibody-mediated effector functions like ADCP and activation of NK cells in humans as well as macaques and rabbits.

The ConM SOSIP.v7 Env trimer was previously shown to induce potent NAbs [17–21]. This study identified that this trimer also induces an array of antibody-mediated effector functions in human vaccine recipients. Induction of these functions appears to be highly coordinated with strong correlations between the different functions. This is a positive outcome, since antibody polyfunctionality was previously shown to be associated with reduced HIV-1 disease progression [32]. There was a strong induction of antibodies able to engage FcγRs and C1q already after two vaccinations and this remained stable after a third vaccination. In contrast, we observed a small additional improvement in the ability of serum antibodies to mediate ADCP and activation of NK cells after the third vaccination, albeit not significant. This may possibly be explained by increased affinity due to sequential boosting.

In previous assessments of antibody titer and neutralization capacity, an interesting difference in the immunogenicity of ConM SOSIP.v7 vaccination between females and males was identified [22]. In the current study, we looked further into this finding and found that antibody-mediated effector functions also differed between females and males. Interestingly, a principal component analysis indicated subtly larger contributions of FcγR and C1q interaction compared to antibody titers, suggesting that the functional difference between females and males is not only explained by a difference in antibody quantity. This is consistent with our previous finding that females had higher levels of ConM-specific antibodies of the IgG1 subclass and lower levels of the IgG4 subclass compared to males [22]. Indeed, IgG1 is associated with induction of effector functions while IgG4 is known to have limited functionality, and this corresponds well with our findings of higher FcγR and C1q interaction in female participants [33]. Others have observed higher antibody titers in females after various antiviral vaccines such as against Influenza virus, yellow fever virus and hepatitis B [34,35]. Differences between antibody Fc glycosylation between females and males could be a potential underlying mechanism [36]. We were unable to investigate whether these sex-specific differences also translate to our animal models due to the absence of male animals. However, others have shown that sex-specific differences do exist in macaques [11,37]. Therefore, it is of interest to include both sexes in future animal studies as well as human studies to more completely assess immunogenicity as representative for the entire population.

To assess the immunogenicity of early-stage vaccine candidates as completely as possible, it is imperative to have suitable models. In this study, we also sought to explore how effective rabbits and macaques predict the antibody response elicited by a HIV-1 Env native-like trimer vaccine in humans. It was previously shown that the ConM SOSIP.v7 protein vaccine induced strong IgG and NAb titers in rabbits and macaques [18–21,38] and more recently in healthy human volunteers [22]. Even though the use of rabbit and macaque models is very common in the HIV-1 field, it is not yet standard

practice to also assess functions beyond antibody titers and neutralization capacity, especially in rabbits. Decisions on which vaccine to move forward in the developmental pipeline are often based on neutralization data in preclinical models. Animal immunogenicity studies rely on the assumption that higher responses in animals translate to higher responses in humans [39]. However, in several instances vaccine candidates with promising results in macaque models did not lead to favorable outcomes in human studies. For example, the Mosaico and Imbokodo clinical studies failed to show efficacy, while a preceding macaque study demonstrated 67% protection from SHIV acquisition [40–42]. While the vaccine tested in the Mosaico en Imbokodo trials did not induce NAbs, the trials did expose a need for more information to improve interpretation of preclinical results, including information on antibody-mediated effector functions [43,44]. When we compared the different functionalities of the antibody response in the clinical trial with the responses to the same vaccine in rabbits and macaques, we observed fairly similar responses, with a few quantitative differences, most notably higher activation of NK cells by rabbit sera. In macaques, we did observe a slightly weaker ability of serum antibodies to interact with cynomolgus macaque FcγRIIa, associated with ADCP, and with human C1q, associated with complement activation. Despite the lower ratio between FcγRIIa interaction and IgG levels in the macaques, ADCP scores were similar to the human study, indicating that these two assays have a different profile for predicting responses between specifies. While we were not able to work with macaque target or effector cells in our cell-based assays, we were able to generate FcγR reagents specific for cynomolgus macaques. Assays based on FcγR and C1q interaction compared to assays based on the activation of immune cells to perform antiviral functions, likely bring different value as methodology to predict antibody functionality between species. While cell-based assays present a more biological approximation, interaction-based assays are generally more robust, consistent and comparable.

There are many factors underlying potency of antibody-mediated effector functions. Distribution of antibody (sub) classes, antibody glycosylation, epitope specificity and antibody valency all are known to affect antibody Fc functionality and could be further investigated to gain additional insight on how these functions are induced in each model [45]. Furthermore, it is known that the macaque inhibitory FcγRIIb receptor has higher affinity for antibodies than its human counterpart [23]. The possibility that this could play a role by dampening the effector functions may also be of interest to research in future studies. Moreover, the functional differences between human and macaque IgG subclasses limit the relevance of studying subclass distribution in macaques, e.g., by lack of a counterpart for the polyfunctional IgG3 subclass in humans. The rabbit humoral immune system is also suitable to study antibody-mediated effector functions and this compatibility has been utilized before [25–27]. While no methods have been developed yet for studying interaction with rabbit-derived immune proteins, rabbit IgG was shown to be compatible with human C1q as well as FcγRIIa and FcγRIIIa on a structural level, with largely identical amino acid residues responsible for these interactions [46]. We observed that rabbits generated similar levels of antibodies interacting with human FcγR and C1q protein as the clinical trial participants. Others have found that after HIV-1 vaccination, rabbit sera had higher levels of ADCC, activation of NK cells (measured by CD107 expression) and engagement with FcγRIIIa[25] compared to macaques. We did not find a higher ability of rabbit sera to interact with FcγRIIIa, however the increased ability to activate NK cells is consistent with our findings. Species differences could potentially influence the ratio of targeted epitopes and thereby influence the level of effector functions. In addition, rabbit antibodies were shown to have much lower levels of fucosylation [47]. Since afucosylated antibodies are known to be more potent activators of NK cells [48], this could explain why we see much higher NK cell activation in the rabbits. We were limited by the lack of available methodology using species-matched effector cells and it would be interesting to see if species-matched assays lead to higher responses in future studies. When we compared neutralizing antibody responses between the three models, we saw that rabbit sera had higher neutralizing capacity while macaque sera had slightly lower neutralizing capacity. Neither of the observed differences should prevent implementation of these assays in preclinical studies. Rather, they demonstrate the possibility of establishing the species-specific differences to allow prediction of human responses based on either animal model. When establishing these differences, the subspecies of animals should also be taken into account: we studied cynomolgus macaques and there may be differences if we would have used another subspecies.

We observed a lower immune response in the s.c. vaccinated macaques compared to the i.m. vaccinated macaques. In addition, there were large differences between the different s.c. immunized animals, indicating that this administration method is less effective. Adjuvants are also known to affect the profile of an immune response. We compared MPLA liposomes with oil-in-water squalene emulsion as adjuvants. MPLA is a TLR4 agonist and this liposomal formulation has shown promise in several vaccine trials [49]. Squalene Emulsion is similar to the adjuvant MF59, which is used in seasonal influenza vaccines [50]. In the macaque model, we observed a small, non-significant increase in antibody level and functionality when squalene emulsion was used. In the rabbits, this difference was only visible after two vaccinations and not after three. The PCA of the responses in macaques also revealed a small difference between the two adjuvants while this was not noticeable for the rabbits. Unfortunately, we were not able to also make this comparison in the human study since there were no differences in adjuvant type. However, a human study with a HIV-1 gp120 vaccine previously also showed higher antibody responses and functionality when the antigen was formulated with squalene emulsion compared to MPLA liposomes [51]. An ongoing clinical study in which the BG505 SOSIP.664 native-like trimer is formulated with different adjuvants will provide new insights (ClinicalTrials.gov identifier NCT04177355). There are likely differences in how adjuvants work in different animal models, for example because of differences in the distribution and function of innate immune receptors, which are vital for the mechanism of action for adjuvants. Therefore, although representative animal models can be informative in early stages of vaccine development, human studies are eventually needed to define the utility of a given adjuvant. Interestingly, others have shown for Influenza that adjuvants may also affect the breadth of the antibody response, which would be very desirable for HIV-1 and should be part of future studies [52]. In both macaque and rabbit models, the largest differences between groups and between animals were observed when investigating C1q interaction, compared to FcγRIIa and FcγRIIIa interaction. This is likely influenced by avidity, as C1q binds optimally when six Fc tails are in close proximity [53]. In contrast, dimerization is required for FcγRs and a minimum of two tails in close proximity is sufficient [54].

In conclusion, we established that vaccination with HIV-1 Env native-like trimers elicits antibodies able to induce effector functions in humans. Various antibody functions were induced and this functionality was closely coordinated. Moreover, more potent antibody Fc functions were elicited in female compared to male participants. We also showed similar induction of these functions in macaque and rabbit models, with a few differences that may be taken into account when translating findings from preclinical studies to humans. Our study also reveals several limitations, such as the availability of rabbit-specific reagents and methodology using species-specific cells, which are of interest to further augment the quality and biological relevance of information that can be obtained in future preclinical studies. Furthermore, we found that changes in adjuvant or administration route of vaccines can alter the Fc functionality of vaccine-elicited antibodies. The elicitation of antibody-mediated effector functions in combination with neutralizing capacity could provide a benefit towards an eventual efficacious vaccine. These findings should guide interpretation of preclinical HIV-1 vaccine studies and inform the selection of vaccination regimens for clinical studies.

## Materials and methods

### Ethics statement

The ACTHIVE-001 study received approval from the Central Committee on Research Involving Human Subjects, the Ministry of Health, Welfare and Sports of the Netherlands and the Medical Research Ethics Committee of the Amsterdam University Medical Centers (previously 'Academic Medical Center')(file number NL69161.000.19). All participants provided written informed consent. Cynomolgus macaques were housed in compliance with European Directive 2010/63/EU, the French regulations, and the Standards for Humane Care and Use of Laboratory Animals, of the Office for Laboratory Animal Welfare (OLAW, assurance number #A5826-01, US). The protocols were approved by the institutional ethical committee "Comité d'Ethique en Expérimentation Animale du Commissariat à l'Energie Atomique et aux Energies Alternatives » (CEtEA #44) under statement number A15-073. The study was authorized by the "Research, Innovation and Education Ministry" under registration number APAFIS#3132–2015121014521340.

Rabbit experiments were performed under subcontract at the National Food Chain Safety Office, Directorate of Veterinary Medicinal Products (NFCSO-DVMP, Budapest, Hungary). All procedures were approved by the animal ethics committee of NFCSO-DVMP.

### ConM SOSIP.v7 vaccine clinical trial (ACTHIVE-001) in healthy volunteers

Twenty-four healthy, HIV-uninfected, adult volunteers were vaccinated with ConM SOSIP.v7 gp140, adjuvanted with the TLR4 agonist monophosphoryl lipid A (MPLA) as part of the randomized, open-label, uncontrolled phase 1 ACTHIVE-001 study (ClinicalTrials.gov identifier NCT03961438). In the per-protocol cohort, 13 participants received 100 µg ConM SOSIP.v7 gp140 at weeks 0, 8 and 24, while ten participants received 100 µg ConM SOSIP.v7 gp140 at weeks 0 and 8 and 20 µg ConM SOSIP.v7 gp140 at week 24. All vaccinations were adjuvanted with a consistent dose of 500 µg MPLA liposomes and administered intramuscularly (i.m.) in the deltoid muscle of the non-dominant arm. Participants were followed up for safety and immunogenicity evaluation until one year after the third vaccination. PBMC, plasma and serum collection was performed on, but not limited to, the day of each vaccination and one and two weeks thereafter. In the current study, serum samples were analyzed at baseline (week 0) and two weeks after the second and third vaccination (weeks 10 and 26).

### ConM SOSIP.v7 vaccine study in cynomolgus macaques

Three groups of six cynomolgus macaques (*Macaca fascicularis*) originating from Mauritius and imported from AAA-LAC certified breeding centers were housed in IDMIT infrastructure facilities (CEA, Fontenay-aux-roses, France, Animal facility authorization #D92-032–02). All animals were female and aged between 33–42 months. They were immunized with pre-fusion ConM SOSIP.v7 HIV-1 envelope trimers, either with MPLA liposomes delivered i.m., with MPLA liposomes delivered subcutaneously (s.c.) or with squalene emulsion as adjuvant and delivered i.m.. Immunizations were performed at weeks 0, 8 and 24 with 20 µg ConM SOSIP.v7. The serum samples for this study were collected one week prior to the first immunization and two weeks after each immunization (weeks 2, 10 and 26).

### ConM SOSIP.v7 vaccine study in rabbits

The rabbits were female, New Zealand white rabbits. Three groups of six female New Zealand white rabbits were immunized with ConM SOSIP.v7. One group received no adjuvant, one group received ConM SOSIP.v7 formulated with 1:1 volume per volume squalene emulsion adjuvant and one group received ConM SOSIP.v7 formulated with 50 µL MPLA liposomes adjuvant. Injections were given intramuscularly at week 0, 4 and 20 containing 20 µg ConM SOSIP.v7. The serum samples for this study were collected at week 0, 2, 6 and 22 after the first immunization.

### Env protein production

Native-like SOSIP stabilized pre-fusion HIV envelope glycoprotein trimers were described previously [17]. Env trimers were produced in HEK293F cells (Invitrogen), cultured in Freestyle medium (Life Technologies) and transfected with polyethylenimine hydrochloride (PEI) MAX (Polysciences) at 1 mg/L and plasmids containing the envelope gene and furin (4:1 ratio, 312.5 µg/L in total) in 50 mL OptiMEM (Gibco) per liter for seven days. Protein was then purified from the 0.22 µM filtered supernatant using CNBr-activated Sepharose 4B (GE Healthcare) affinity columns conjugated with PGT145. Next, proteins were further purified using size exclusion chromatography on a Superdex200 10/330 G/L column (GE healthcare) to purify the trimer fraction only. Concentration steps were performed with Vivaspin centrifugal concentrators (Sartorius).

### Production of human Fc gamma receptor ectodomain dimers

Ectodomain dimers of human FcγRIIa-H131 and FcγRIIIa-V158 (high affinity allelic variants) used to probe ability of antibodies to bind these receptors were produced in HEK293F cells maintained in FreeStyle medium. Plasmids were

designed by Bruce Wines and Mark Hogarth of the Burnet Institute in Melbourne, Australia and contained the genes followed by a hexahistidine (His)-tag and Avi-tag. Transfections were performed using PEI MAX at 1 mg/L and the plasmids at 312.5 µg/L in 50 mL OptiMEM per liter. Seven days post transfection the protein was purified from the 0.22 µM filtered supernatant with affinity chromatography using Nickel-Nitrilotriacetic Acid (NiNTA) agarose beads (QIAGEN). Eluates were concentrated and buffer exchanged to PBS using 30 kDa molecular weight cutoff (MWCO) Vivaspin centrifugal concentrators (Sartorius). Protein was labeled with biotin on the Avi-tag with birA ligase (Genecopoeia). Protein was further purified by size exclusion chromatography using a SuperDex200 10/300 GL increase column to purify the non-aggregated, intact protein only (Vivaspin centrifugal concentrators).

## Generating cynomolgus macaque Fc gamma receptor ectodomain dimers

Pig-tailed macaque (Macaca Nemestrina) Fc gamma receptor ectodomain dimer sequences with His- and Avi-tag were generously provided by Bruce Wines and Mark Hogarth of the Burnet Institute (Melbourne, Australia). Three amino acids were identified to be different between pigtail and cynomolgus macaques for the FcγRIIa ectodomain gene. The amino acid sequences of the pig-tailed and cynomolgus macaque FcγRIIIa ectodomain were identified to be identical. The pig-tailed macaque FcγRIIa ectodomain dimer sequence was converted into a cynomolgus macaque ectodomain dimer sequence and the genes were synthesized by IDT technologies. Each gene was inserted into an expression vector containing a His- and Avi-Tag. The proteins were produced with the exact same procedure as the human Fc gamma receptor ectodomain dimers.

## Coupling of HIV protein to microspheres

Luminex Magplex beads were coupled using a two-step carbodiimide reaction with 1-Ethyl-3-(3-dimethylaminopropyl) carbodiimide (EDC, Thermo Scientific) and Sulfo-N-Hydroxysulfosuccinimide (Sulfo-NHS, Thermo Scientific) at a ratio of 75 µg SOSIP trimer to 12,5 million beads. All incubations were performed at room temperature on a rotator. Beads were activated with EDC and Sulfo-NHS Beads for 30 minutes, then beads and antigen were incubated together for three hours. Next, the beads were blocked for 30 minutes with PBS containing 2% BSA, 3% fetal calf serum and 0.02% Tween-20 at pH 7.0. Finally, the beads were stored in in PBS containing 0.05% sodium azide at 4 °C until use. Beads were subjected to the same procedure but without addition of any protein as a negative control.

## Luminex assay

15 beads per µL were incubated in a 1:1 ratio with diluted serum overnight at 4 °C. In prior optimization experiments, sera were titrated to determine the optimal dilution factor for each type of sera and each assay. This resulted in a dilution factor of 1:100,000 (human and rabbit samples) and 1:10,000 (macaque samples) for IgG, 1:500 (human samples) 1:5000 (rabbit samples) and 1:100 (macaque samples) for FcγRIIa and FcγRIIIa and 1:500 (human and rabbit samples) and 1:200 (macaque samples) for C1q. The next day, plates were washed. Detection of IgG was performed with goat anti-human IgG-PE (Southern Biotech) for two hours, mouse anti-rabbit IgG-PE (Southern Biotech) for two hours or goat anti-monkey IgG-biotin (Sigma Aldrich) for two hours followed by Streptavidin-PE (Invitrogen) for one hour. For human and rabbit sera, biotinylated ectodomain dimers of human FcγRIIa-H131 and FcγRIIIa-V158 were incubated two hours followed by one hour of Streptavidin-PE, while for cynomolgus macaque sera, biotinylated cynomolgus macaque-specific ectodomain dimers of FcγRIIa and FcγRIIIa were incubated for two hours, followed by Streptavidin-PE for one hour. For all sera, purified human C1q was biotinylated with the EZ-Link Sulfo-NHS-LC-Biotinylation Kit (Thermo Fisher Scientific) following the manufacturer's protocol, conjugated to Streptavidin-PE and incubated 30 minutes at room temperature before use in the assays and in the assay it was incubated for two hours. Finally, plates were washed and beads were resuspended in drive fluid (Luminex). Read-out was performed on a Magpix instrument (Luminex). Resulting median fluorescence intensity (MFI) values were corrected by subtraction of MFI values from buffer and beads only wells. A cut-off

for a detectable response was set as follows: a response above 100 MFI, more than 3x the MFI of the no protein control bead in the same well and above the 95th percentile of the measurements of all individuals/animals in the same group at baseline. The assay for all samples after the third vaccination was repeated at equal dilution for each human, macaque and rabbit sera to allow better comparability, using 1:100,000 for IgG, 1:1,000 for FcγRIIa and FcγRIIIa and 1:500 for C1q.

### ADCP assay

ADCP assays were performed similar to previously described [55]. Fluorescent Neutravidin beads (Invitrogen) were incubated with biotinylated ConM SOSIP.v7 protein overnight at 4 °C. Beads were spun down and washed twice in PBS + 2% BSA. The coated beads were resuspended in PBS + 2%BSA and 0.1 µl of the original suspension was placed in every well of a V-bottom 96 well plate and incubated (two hours at 37 °C) with 1:500 diluted serum. After incubation, plates were washed and $5 \times 10^4$ THP-1 effector cells (ATCC) were added to each well in a final volume of 100 µl. Subsequently, plates were shortly spun down to promote bead-to-cell contact before incubation (five hours at 37°C). After incubation, the cells were washed, resuspended in PBS + 2% FCS and analyzed by flow cytometry on a FACS Canto (BD Biosciences). Phagocytic activity was determined by the amount of cells containing FITC labeled beads x the FITC mean fluorescence intensity.

### NK cell activation

NK cell activation assays were performed similar to previously described [56]. PBMC were obtained from leukapheresis products of healthy donors by Ficoll-Paque density gradient following the manufacturer's protocol. NK cells were enriched from human PBMCs by negative selection using a human NK Cell Isolation Kit (Miltenyi) following manufacturer's protocol and stimulated with 10 ng/mL IL-15 in Iscove's Modified Dulbecco's Medium (Gibco) supplemented with 10% FCS and 100 U/mL penicillin/streptomycin. Stimulated NK cells were incubated at 37 °C overnight. NiNTA ELISA plates (QIAGEN) were coated with ConM SOSIP.v7 (2,5 µg/ml) in 1xTBS overnight at 4°C. Plates were washed 3x with 1x TBS, blocked with PBS-1% BSA for one hour at 37 °C and thereafter washed 5x with TBS. Plates were incubated with 1:100 diluted serum in PBS-1% BSA for two hours at 37 °C. After incubation, the plates were washed five times with TBS and then incubated with 50 µl stimulated NK cells ($5 \times 10^4$ cells/well) supplemented with 10 ng/mL IL-15 and 1:3000 Brefeldin-A. As a positive control, phorbol myristate acetate (PMA; 50 ng/mL) and ionomycin (500 ng/mL) were added to some wells. The plate was incubated three hours at 37 °C. Then, NK cells were transferred to a 96-well V-bottom plate, washed using FACS buffer (2% FCS, PBS) and stained with anti-CD16-PE (Biolegend), anti-CD107a-APC (Biolegend) and Fixable Viability Dye eFluor 780 (eBioscience) for 30 minutes at 4 °C. Cells were fixated with the Cytofix/Cytoperm Fixation/Permeabilization Kit (BD Biosciences) for 20 minutes at 4 °C and then stored at 4 °C in FACS buffer overnight. The next day, NK cells were stained with PE anti-human IFN-γ clone B27 (BioLegend). Cells were washed and resuspended in FACS buffer, and then analyzed using the FACS Symphony A1 (BD), gated for live cells and percentages of CD107, CD16 and IFN-γ positive NK-cells were determined. This assay was previously shown to correlate well with assays using HIV-1 infected cells [56].

### Neutralization assays

TZM-bl cell neutralization assays were performed using Env-pseudotyped viruses as previously described [22,57]. Briefly, 3-fold serial dilutions of heat-inactivated serum samples were incubated in duplicate with a pre-titrated dose of virus in a total volume of 150 µl in 96-well flat-bottom culture plates, for one hour at 37 °C. 10,000 TZM-bl cells were added to each well in 75 µl of growth medium (GM) containing 45 µg/mL DEAE dextran. The medium was removed after 48 hours and 50 µl of Bright-Glo reagent (Promega, Madison, Wisconsin, USA) diluted 1:2 with GM was added to each well. Plates were incubated at room temperature for two minutes to allow complete cell lysis. 40 µl was transferred to white 96-well plates and analyzed in a luminometer (Mithras (Berthold, Germany)). Neutralization titers were calculated as the serum dilution

where relative luminescence units (RLU) were reduced by 50% compared to virus control wells after subtraction of background RLUs.

## Data analysis and statistics

Mann-Whitney U tests were used for comparing two independent groups. A Kruskal-Wallis test followed by a Dunn's multiple comparison test was used for comparing three independent groups. A Friedman test followed by a Dunn's multiple comparison test was used for comparing three time points for the same individuals. Correlation analysis were done with Spearman's rank correlation. All statistical analyses were performed in GraphPad Prism v9.5. Principal Component Analyses were performed in Matlab R2022b and PLS Toolkit (Eigenvector).

## Supporting information

**S1 Fig. Spearman correlations between different measurements of antibody level and functionality after ConM SOSIP.v7 vaccination in healthy volunteers.** Spearman correlations between ConM-specific IgG levels and interaction of ConM-specific antibodies with (A) FcγRIIa, (B) FcγRIIIa and (C) C1q. (D) Spearman correlations between ConM-specific IgG levels and half-maximal infective dilution ($ID_{50}$) neutralization titer against ConM pseudovirus. (E) Spearman correlations between interaction of ConM-specific antibodies with human FcγRIIa and antibody dependent cellular phagocytosis (ADCP) of ConM SOSIP.v7 conjugated beads by THP-1 cells. The ADCP score is composed of the mean fluorescence intensity multiplied by the count of internalized beads. (F) Spearman correlations between interaction of ConM-specific antibodies with human FcγRIIIa and activation of NK cells derived from healthy donor PBMCs by ConM-specific serum antibodies. The NK cell activation is presented as the percentage of cells expressing CD107 and IFN-γ. Spearman rho correlation coefficients (r) and P-values are indicated separately for the serum collected at week 10 and week 26. All IgG, FcγRIIa, FcγRIIIa and C1q data shown are median fluoresence intensity (MFI) measured by Luminex assay. Female participants are shown as circles and males as triangles.
(TIF)

**S2 Fig. Comparison of clinical trial participants with female and male sex at birth.** (A) Male and female participants were compared at baseline, after two vaccinations and after three vaccinations for IgG levels, (B) interaction of ConM-specific antibodies with human FcγRIIa, (C) human FcγRIIIa and (D) human C1q. All measurements were performed by Luminex immunoassay and presented as blank-corrected median fluoresence intensity (MFI). (E) Antibody dependent cellular phagocytosis (ADCP) of ConM SOSIP.v7 conjugated beads by THP-1 cells mediated by serum antibodies compared between male and female clinical trial participants. The ADCP score is composed of the mean fluorescence intensity multiplied by the count of internalized beads. (F) NK cell activation by ConM-specific serum antibodies, presented as the percentage of cells expressing CD107 and IFN-γ. (G) $ID_{50}$ neutralization titer against ConM pseudovirus compared between clinical trial participants with a full or fractional third dose. Groups were compared at each time point with a Mann-Whitney U test. *=P<0.05, **=P<0.01, ***=P<0.001.
(TIF)

**S3 Fig. Principal Component Analysis of serum responses in clinical trial participants with female and male sex at birth and normalized ADCP and NK cell activation responses.** (A) Principal component analysis (PCA) over all data presented in the graphs in Fig 2 to identify the features that differ the most between female and male participants. Axes of the graph indicate the percentage of data variance explained by the first two principal components (PC). (B) Loading bar charts explaining the contribution of the different measurements to the separation of the groups on PC 1 and PC 2. (C) PCA for serum responses after two vaccinations (week 10) to identify the features that differ most between male and female participants. (D) Loading bar charts explaining the contribution of the different ConM-specific measurements to the separation of the groups on PC 1 and PC 2. (E) ADCP and (F) NK cell activation responses at week 10, normalized by

IgG levels. Week 10 IgG levels were log-transformed and then a normalization factor was calculated as the fold-change in log IgG titers compared to the average log IgG titer. Results for ADCP and NK cell activation where multiplied by this factor to obtain the normalized values. *=P<0.05.
(TIF)

**S4 Fig. ConM-specific IgG levels and interaction of serum antibodies with FcγRIIa, FcγRIIIa and C1q compared between ConM vaccinated clinical trial participants, macaques and rabbits.** (A) ConM-specific IgG levels expressed as median fluorescence intensity (MFI) measured by Luminex assay after three vaccinations (week 26 for macaques and the human study, week 22 for rabbits). Species-specific secondary antibodies were used. (B) Interaction of ConM-specific antibodies with FcγRIIa expressed as MFI measured by Luminex assay after three vaccinations. Human FcγRIIa was used for the human and rabbit study while cynomolgus macaque FcγRIIa was used for the macaque study. (C) Interaction of ConM-specific antibodies with FcγRIIIa expressed as MFI measured by Luminex assay after three vaccinationsHuman FcγRIIIa was used for the human and rabbit study while cynomolgus macaque FcγRIIIa was used for the macaque study. (D) Interaction of ConM-specific antibodies with C1q expressed as MFI measured by Luminex assay after three vaccinations. Human C1q was used for all three studies. Sera of each species were diluted equally to facilitate these comparisons. Since the data comprised different species, the results were assessed qualitatively and without statistical analysis. Monoclonal antibodies with known effector function activity included in the ADCP and NK cell activation assays as controls. The monoclonal antibodies were tested at a concentration of 1 µg/mL and the sera shown in Fig 3 were tested at a dilution of 1:500. (G) Overlay of the three spider plots shown in Fig 3.
(TIF)

**S5 Fig. Comparisons of ConM-specific IgG levels and antibody functionality between cynomolgus macaques vaccinated with ConM SOSIP.v7 using different administration routes and adjuvants at additional time points.** (A) ConM-specific IgG levels in serum before vaccination (week -1), after one vaccination (week 2), after two vaccinations (week 10) and after three vaccinations (week 26). (B) Interaction of ConM-specific antibodies with cynomolgus macaque FcγRIIa. (C) Interaction of ConM-specific antibodies with cynomolgus macaque FcγRIIIa. (D) Interaction of ConM-specific antibodies with human C1q. Data for panels A-D was measured by Luminex immunoassay and is presented as blank-corrected median fluoresence intensity (MFI). (E) Antibody dependent cellular phagocytosis (ADCP) of ConM SOSIP.v7 conjugated beads by THP-1 cells mediated by serum antibodies at baseline (week -1) and after three vaccinations (week 26). The ADCP score is composed of the mean fluorescence intensity multiplied by the count of internalized beads. (F) NK cell activation by ConM-specific serum antibodies before vaccination (week -1) and after three vaccinations (week 26), presented as the percentage of cells expressing CD107 and IFN-γ. At each timepoint, all three groups were compared with each other using a Kruskal-Wallis test followed by a Dunn's multiple comparison test. *=P<0.05, **=P<0.01.
(TIF)

**S6 Fig. Spearman correlations between different measurements of antibody level and functionality after three ConM SOSIP.v7 vaccinations in cynomolgus macaques.** Spearman correlations between ConM-specific IgG levels and interaction of ConM-specific antibodies with (A) cynomolgus macaque FcγRIIa, (B) cynomolgus macaque FcγRIIIa and (C) human C1q. (D) Spearman correlations between ConM-specific IgG levels and half-maximal infective dilution ($ID_{50}$) neutralization titer against ConM pseudovirus. (E) Spearman correlations between interaction of ConM-specific antibodies with cynomolgus macaque FcγRIIa and antibody dependent cellular phagocytosis (ADCP) of ConM SOSIP.v7 conjugated beads by THP-1 cells mediated by serum antibodies. The ADCP score is composed of the mean fluorescence intensity multiplied by the count of internalized beads. (F) Spearman correlations between interaction of ConM-specific antibodies with cynomolgus macaque FcγRIIIa and NK cell activation by ConM-specific serum antibodies, presented as the percentage of cells expressing CD107 and IFN-γ. Spearman rho (r) correlation coefficients and P-values are indicated

on each graph. All IgG, FcγRIIa, FcγRIIIa and C1q results shown are median fluorescence intensity (MFI) measured by Luminex assay on serum collected two weeks after the third vaccination (week 26). (G) Principal component analysis (PCA) over all data presented in the graphs of Fig 4 to identify features that differ the most between experimental groups. Axes of the graph indicate the percentage of data variance explained by the first two principal components (PC). (H) Loading bar charts explaining the contribution of the different measurements to the separation of the groups on PC 1 and PC 2. (TIF)

**S7 Fig. Normalized ADCP and NK cell activation responses after three ConM SOSIP.v7 vaccinations in cynomolgus macaques and FcγRIIa, FcγRIIIa to heterologous HIV-1 antigens.** (A) ADCP and (B) NK cell activation responses at week 26, normalized by IgG levels. Week 26 IgG levels were log-transformed and then a normalization factor was calculated as the fold-change in log IgG titers compared to the average log IgG titer. Results for ADCP and NK cell activation where multiplied by this factor to obtain the normalized values. (C) The intraction of antibodies binding the heterologous antigens ConS SOSIP and (D) clade A BG505 SOSIP with FcγRIIa. (E) The intraction of antibodies binding the heterologous antigens ConS SOSIP and (F) clade A BG505 SOSIP with FcγRIIIa. Data for panels C-F was measured by Luminex immunoassay and is presented as blank-corrected median fluoresence intensity (MFI). In each graph, all three groups were compared with each other using a Kruskal-Wallis test followed by a Dunn's multiple comparison test. *=P<0.05. (TIF)

**S8 Fig. Comparisons of ConM-specific IgG levels and antibody functionality between rabbits vaccinated with ConM SOSIP.v7 using no adjuvant, MPLA or squalene emulsion at additional time points.** (A) ConM-specific IgG levels in serum before vaccination (week 0), after one vaccination (week 2), after two vaccinations (week 6) and after three vaccinations (week 22). (B) Interaction of ConM-specific antibodies with human FcγRIIa at the same time points. (C) Interaction of ConM-specific antibodies with human FcγRIIIa. (D) Interaction of ConM-specific antibodies with human C1q. Data for panels A-D was measured by Luminex immunoassay and presented as blank-corrected median fluorescence intensity (MFI). (E) Antibody dependent cellular phagocytosis (ADCP) of ConM SOSIP.v7 conjugated beads by THP-1 cells mediated by serum antibodies at baseline (week 0) and after three vaccinations (week 22). The ADCP score is composed of the mean fluorescence intensity multiplied by the count of internalized beads. (F) NK cell activation by ConM-specific serum antibodies before vaccination (week 0) and after three vaccinations (week 22), presented as the percentage of cells expressing CD107 and IFN-γ. At each timepoint, all three groups were compared with each other using a Kruskal-Wallis test followed by a Dunn's multiple comparison test. *=P<0.05, **=P<0.01, ***=P<0.001. (TIF)

**S9 Fig. Spearman correlations between different measurements of antibody level and functionality after three ConM SOSIP.v7 vaccinations in rabbits.** Spearman correlations between ConM-specific IgG levels and interaction of ConM-specific antibodies with (A) human FcγRIIa, (B) human FcγRIIIa and (C) human C1q. (D) Spearman correlations between ConM-specific IgG levels and half-maximal infective dilution ($ID_{50}$) neutralization titer against ConM pseudovirus. (E) Spearman correlations between interaction of ConM-specific antibodies with human FcγRIIa and antibody dependent cellular phagocytosis (ADCP) of ConM SOSIP.v7 conjugated beads by THP-1 cells mediated by serum antibodies. The ADCP score is composed of the mean fluorescence intensity multiplied by the count of internalized beads. (F) Spearman correlations between interaction of ConM-specific antibodies with human FcγRIIIa and NK cell activation by ConM-specific serum antibodies, presented as the percentage of cells expressing CD107 and IFN-γ. All IgG, FcγRIIa, FcγRIIIa and C1q results shown are median fluoresence intensity (MFI) measured by Luminex assay on serum collected two weeks after the third vaccination (week 22). (G) Correlations analysis combining ADCP scores and neutralization titers at two weeks after the third vaccination for all three species (human, macaque and rabbit). (H) Correlations analysis combining the percentage of activated NK cells and neutralization titers at two weeks after the third vaccination for all three species (human, macaque and rabbit). Only human subjects that received the full dose regimen (N=13) were incuded in the analysis for

panels G and H. Spearman rho (r) correlation coefficients and P-values are indicated on each graph. (I) Principal component analysis (PCA) over all data presented in the graphs of Fig 5 to identify the features that differ the most between experimental groups. Axes of the graph indicate the percentage of data variance explained by the first two principal components (PC). (J) Loading bar charts explaining the contribution of the different measurements to the separation of the groups on PC 1 and PC 2.
(TIF)

**S1 File. All raw data.** Excel file containing all data included in the figures belonging to this manuscript.
(XLSX)

## Acknowledgments

We thank all clinical trial participants and study staff. We thank Bruce Wines and Mark Hogarth of the Burnet Institute for the design of the pig-tailed macaque specific Fc gamma receptor ectodomain dimers (used to generate cynomolgus macaque specific versions) and the human Fc gamma receptor ectodomain dimers; Mariangela Cavarelli of the CEA for providing plasmids and Amy Chung of the University of Melbourne for guidance, training and advice. We thank the staff of the animal facility of IDMIT, particularly B. Delache, M. Pottier, S. Langlois, J.M. Robert, N. Dhooge, and R. Ho Tsong Fang. Dietmar Katinger provided the adjuvants for the NHP study.

## Author contributions

**Conceptualization:** Marloes Grobben, Rogier W. Sanders, Marit J. van Gils.

**Formal analysis:** Marloes Grobben.

**Funding acquisition:** Rogier W. Sanders, Marit J. van Gils.

**Investigation:** Marloes Grobben, Alex Rooker, Chunhao Yu, Khadija Tejjani, Monica Tolazzi.

**Methodology:** Angela I. Schriek, Steven W. de Taeye.

**Resources:** Emma I. M. M. Reiss, Angela I. Schriek, Karlijn van der Straten, Nathalie Dereuddre-Bosquet, Pauline Maisonnasse, Réka Felfödiné Lévai, Attila Farsang, Roger Le Grand, Gabriella Scarlatti, Steven W. de Taeye, Godelieve J. de Bree.

**Supervision:** Rogier W. Sanders, Marit J. van Gils.

**Validation:** Marloes Grobben.

**Visualization:** Marloes Grobben.

**Writing – original draft:** Marloes Grobben, Marit J. van Gils.

**Writing – review & editing:** Marloes Grobben, Emma I. M. M. Reiss, Angela I. Schriek, Karlijn van der Straten, Nathalie Dereuddre-Bosquet, Pauline Maisonnasse, Alex Rooker, Chunhao Yu, Khadija Tejjani, Monica Tolazzi, Kwinten Sliepen, Réka Felfödiné Lévai, Attila Farsang, Roger Le Grand, Gabriella Scarlatti, Robin J. Shattock, Steven W. de Taeye, Godelieve J. de Bree, Rogier W. Sanders, Marit J. van Gils.

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
