## [Decision Letter · Decision Letter 0]

10 Mar 2025

PPATHOGENS-D-25-00148

Induction of HIV-1-specific antibody-mediated effector functions by native-like envelope trimers in humans

PLOS Pathogens

Dear Dr. Grobben,

Thank you for submitting your manuscript to PLOS Pathogens. After careful consideration, we feel that it has merit but does not fully meet PLOS Pathogens's publication criteria as it currently stands. Therefore, we invite you to submit a revised version of the manuscript that addresses the points raised during the review process.

Please submit your revised manuscript within 60 days May 09 2025 11:59PM. If you will need more time than this to complete your revisions, please reply to this message or contact the journal office at plospathogens@plos.org. Please include the following items when submitting your revised manuscript:

We look forward to receiving your revised manuscript.

Kind regards,

Penny L. Moore

Academic Editor

PLOS Pathogens

Susan Ross

Section Editor

PLOS Pathogens

Sumita Bhaduri-McIntosh

Editor-in-Chief

PLOS Pathogens

orcid.org/0000-0003-2946-9497

Michael Malim

Editor-in-Chief

PLOS Pathogens

orcid.org/0000-0002-7699-2064

**Journal Requirements:**

1) We do not publish any copyright or trademark symbols that usually accompany proprietary names, eg ©,  ®, or TM  (e.g. next to drug or reagent names). Therefore please remove all instances of trademark/copyright symbols throughout the text, including:

- TM on page: 25.

3) We notice that your supplementary Figures are included in the manuscript file. Please remove them and upload them with the file type 'Supporting Information'. Please ensure that each Supporting Information file has a legend listed in the manuscript after the references list.

Potential Copyright Issues:

i) Figures 1A, 2A, 4A, and 5A. Please confirm whether you drew the images / clip-art within the figure panels by hand. If you did not draw the images, please provide (a) a link to the source of the images or icons and their license / terms of use; or (b) written permission from the copyright holder to publish the images or icons under our CC BY 4.0 license. Alternatively, you may replace the images with open source alternatives. See these open source resources you may use to replace images / clip-art:

5) We note that your Data Availability Statement is currently as follows: "All relevant data are within the manuscript and its Supporting Information files.". Please confirm at this time whether or not your submission contains all raw data required to replicate the results of your study. Authors must share the “minimal data set” for their submission. PLOS defines the minimal data set to consist of the data required to replicate all study findings reported in the article, as well as related metadata and methods (https://journals.plos.org/plosone/s/data-availability#loc-minimal-data-set-definition).

1) State what role the funders took in the study. If the funders had no role in your study, please state: "The funders had no role in study design, data collection and analysis, decision to publish, or preparation of the manuscript.".

**Reviewers' Comments:**

Reviewer's Responses to Questions

**Part I - Summary**

Reviewer #1: This study focuses on the ability of a HIV Env trimer immunogen (ConM SOSIP.v7) to elicit antibodies that mediate effector functions in preclinical and clinical models. Effector functions of antibodies can contribute to the protective capacity of neutralizing antibodies and are therefore important to elicit during vaccination. The authors demonstrate induction of diverse antibody-mediated effector responses in a human study. They also show that modifying the vaccination regimen in preclinical animal models can alter the functionality of the antibody response. The authors also study the translatability of effector function activity between animal and human models. This is important as little is known about antibody-mediated effector functions in rabbits. These findings will be of broad interest to HIV vaccine field and the wider vaccine field in general.

Reviewer #2: This study makes strides in HIV-1 vaccine development by focusing not only on the induction of broadly neutralizing antibodies (bnAbs) using stabilized, native-like envelope glycoprotein trimers. The research demonstrates that these trimers can elicit multiple, coordinated antibody-mediated effector functions in human volunteers during a phase 1 clinical trial. Notably, female participants exhibited substantially greater functional responses compared to males, which could have implications for future vaccine design. The study also attempted to compare preclinical models involving rabbits and non-human primates, something which has nt been done before. However as expressed in this review, these conclusions are perhaps overdrawn. It highlights the impact of different administration methods and adjuvants on antibody responses similar to what is already known, underscoring the need for a comprehensive approach to vaccine development that maximizes both neutralization capacity and diverse effector functions.

Reviewer #3: The manuscript submitted by van Gils and collaborators effectively utilizes samples from both clinical and preclinical testing of a candidate adjuvanted HIV envelope trimer protein vaccine to study antibody-mediated effector functions in humans and relevant animal models. The manuscript is exceptionally well organized and written, and the figures are clear and informative. The major conclusions from the study were that 1) the candidate vaccine induced humoral immune responses capable of diverse effector functions including phagocytosis, NK cell activation, and complement binding. 2) Both the rabbit and rhesus macaque pre-clinical models could effectively predict these responses, and 3) the potency of antibody-effector responses could be modified by changes to the adjuvant or route of administration. Although none of these conclusions are particularly ground-breaking in-and-of themselves as there is substantial evidence for each in existing literature, the strength of the study is its completeness and ability to demonstrate these conclusions using an ideal cross-species study of the same vaccine/regimen. The primary limitation of the study is that all experiments use the vaccine antigen as the target. Overall, the authors demonstrate strong correlation between levels of antibody binding and the different antibody functions, but would this still be true if the functional assays were performed with a more relevant target, such as HIV virus and infected cells? Or what about a non-vaccine matched protein? Additional work should be performed to address this question.

**Part II – Major Issues: Key Experiments Required for Acceptance**

Reviewer #1: (No Response)

Reviewer #2: 1. The detection of IgG titer was species specific and the macaque and human Fc receptors were species specific. It is not clear why the rabbit samples were screened using human receptors? In addition the functional assays performed (ADCC and ADCP) made use of human cell lines, diluting the applicability of comparisons made across species. As such at least an attempt should be made to use primary macaque cells in at least one of the assays to show that these results are comparable. In addition, the conclusions drawn here do not match the data – 3E for example shows a 2-fold increase in ADCC of rabbit sera compared to all others, also it would be helpful to overlay the spider plots.

2. The authors should normalise for IgG titer (this may exclude C1q binding)

3. Are there antibody controls that the authors can include in these analyses to show that these are indeed potent Fc responses?

Reviewer #3: The lack of testing against a more relevant target - HIV virus or infected cells is a substantial limitation. Addressing this would substantially strengthen the conclusions. At minimum it would be helpful to evaluate breadth against a different trimer as this particular antigen was designed to address the problem of HIV env diversity. Limitations of the study should be addressed in the discussion.

**Part III – Minor Issues: Editorial and Data Presentation Modifications**

Reviewer #1: This manuscript covers several different aspects relating to effector functions. I found the introduction slightly disjointed when these different aspects were introduced.

The NK activation assay shows some donors with a low response despite high FcyRIIIa binding of all samples. Does this not suggest that the functional assay is a better measure of the ADCC activity or that the binding assay is over saturated? What is a meaningful level of ADCC activity for protection?

Were the experiments in figure 3 all done with human Fcg receptors? Wouldn’t you expect the different species to react differently? I can see this information in the figure legend but might be helpful to have in the text as well. For panel 3D, it says that sera was diluted equally to facilitate the comparisons but should these be noramlised to the level of trimer binding antibodies to better compare? Are the differences observed in 3D due to binding to the rabbit IgG FcgR or different epitopes being targeted that facilitate effector functions differently?

In figure 4, the authors show that the different immunization schedules give different absolute amounts of trimer reactive antibodies. This seems to reflect the results for the remaining panels suggesting that the different vaccine protocols are impacting the overall magnitude of the response rather than any specifics about the quality of the effector function response. What would happen if the amount of Trimer-reactive antibody was normalised in the experiment to then determine if there was a change in the effector function activity? Has this been tested?

The authors frequently refer to an unpublished study Reiss et al. It would be helpful to link the changes briefly mentioned in the text on isotype of antibodies to the effector function data seen in this paper.

Reviewer #2: 1. While the authors point to the association of Fc effector function with protection in other diseases, it seems a brief summary of what the current status in HIV vaccination is, is missing in the introduction

2. The introduction could benefit from further editing – several points are reiterated multiple times and could be replaced by some background that is HIV specific

3. The authors state: NHPs also have four IgG subclasses, but they are structurally and functionally more similar to each other than the human versions[17] but fail to mention that this a major issue especially in the context of studying very polyfunctional isotypes such as IgG3.

4. Should the rabbit and macaque Fc responses not be tested using primary cells from the animals themselves? Surely one is underestimating the “autologous” Fc functions?

5. In the results the authors state that “No statistically significant differences were observed between the two dosage groups after the third vaccination.” And they mention that they pooled them – it should be clarified that this refers to pooling fractionated and bolus dosing. Are there differences between these in Fc functions? It’s not clear whether they mean there were no differences for Nab responses etc.

6. The authors should verify that the Fc array performed has not maxed out – seems that this is the case for 1B, D and E where a very poor range of responses is noted and why no increase is seen for these conditions after the third dose. As seem to be the case for adcp and adcc – also fc effector arrays do not tell us “to what extent the elicited antibodies could mediate effector function” but rather tell us the extent to which fc receptors are bound.

7. “Antibody functionality can be expected to be closely linked to antibody quantity.” This sentence in the context it is written is also misplaced. The Fc receptor binding affinity is linked to IgG titer as these output are measured in the same way and not accounted for when running plasma. C1q titers less so as they can be mediated by other isotypes.

8. By “autologous serum neutralization” do the authors mean ConM titers?

9. What happens to the ADCP and ADCC results if igg is normalised for? Ie if score is normalised to IgG titer? Are the authors sure that the sex differences are not because of IgG being higher in females? This difference should be accounted for. PCA in fact shows that all of the top four features are all almost equally accounting for the difference – likely because they are colinear.

10. The scale on figure 3B is confusing and should be either amended or explained

11. The PCA plots drawn are not really informative and are illustrated by the differences in univariant data

12. I think more attention should be drawn to the fact that adjuvant boosting depends on the animal model tested

Reviewer #3: Methods for neutralization assay should be included.

PLOS authors have the option to publish the peer review history of their article (what does this mean?). If published, this will include your full peer review and any attached files.

Reviewer #1: No

Reviewer #2: No

Reviewer #3: No

**Figure resubmission:**
---

## [Decision Letter · Decision Letter 1]

21 Aug 2025

PPATHOGENS-D-25-00148R1

Induction of HIV-1-specific antibody-mediated effector functions by native-like envelope trimers in humans

PLOS Pathogens

Dear Dr. Grobben,

Thank you for submitting your manuscript to PLOS Pathogens. After careful consideration, we feel that it has merit but does not fully meet PLOS Pathogens's publication criteria as it currently stands. Therefore, we invite you to submit a revised version of the manuscript that addresses the points raised during the review process.

Specifically, Reviewer 2 feels that the revised manuscript does not adequately address his/her concerns. My suggestion is to carefully address the reviewer's concerns through some minor edits to the discussion. 

Please submit your revised manuscript within 30 days Oct 20 2025 11:59PM. If you will need more time than this to complete your revisions, please reply to this message or contact the journal office at plospathogens@plos.org. Please include the following items when submitting your revised manuscript:

We look forward to receiving your revised manuscript.

Kind regards,

Penny L. Moore

Academic Editor

PLOS Pathogens

Susan Ross

Section Editor

PLOS Pathogens

Sumita Bhaduri-McIntosh

Editor-in-Chief

PLOS Pathogens

orcid.org/0000-0003-2946-9497

Michael Malim

Editor-in-Chief

PLOS Pathogens

orcid.org/0000-0002-7699-2064

**Journal Requirements:**

**Reviewers' Comments:**

Reviewer's Responses to Questions

**Part I - Summary**

Reviewer #2: (No Response)

Reviewer #3: The updated manuscript addresses the prior minor concerns, and is now suitable for editorial consideration.

**Part II – Major Issues: Key Experiments Required for Acceptance**

Reviewer #2: In general the authors have failed to experimentally address my concerns - while the conclusions have been tempered, I still think that the usefulness of such a study with no additional biological information is limited. The authors also state that "macaque cell usage is complex" but receptors are available commercially from these species so several high-throughput assays could be performed.

Reviewer #3: (No Response)

**Part III – Minor Issues: Editorial and Data Presentation Modifications**

Reviewer #2: (No Response)

Reviewer #3: (No Response)

PLOS authors have the option to publish the peer review history of their article (what does this mean?). If published, this will include your full peer review and any attached files.

Reviewer #2: No

Reviewer #3: No

**Figure resubmission:**
---

## [Editor Report · Decision Letter 2]

10 Oct 2025

Dear Ms. Grobben,

We are pleased to inform you that your manuscript 'Induction of HIV-1-specific antibody-mediated effector functions by native-like envelope trimers in humans' has been provisionally accepted for publication in PLOS Pathogens.

Best regards,

Penny L. Moore

Academic Editor

PLOS Pathogens

Susan Ross

Section Editor

PLOS Pathogens

Sumita Bhaduri-McIntosh

Editor-in-Chief

PLOS Pathogens

orcid.org/0000-0003-2946-9497

Michael Malim

Editor-in-Chief

PLOS Pathogens

orcid.org/0000-0002-7699-2064
---

## [Editor Report · Acceptance letter]

Dear Ms. Grobben,

We are delighted to inform you that your manuscript, "Induction of HIV-1-specific antibody-mediated effector functions by native-like envelope trimers in humans," has been formally accepted for publication in PLOS Pathogens.

Best regards,

Sumita Bhaduri-McIntosh

Editor-in-Chief

PLOS Pathogens

orcid.org/0000-0003-2946-9497

Michael Malim

Editor-in-Chief

PLOS Pathogens

orcid.org/0000-0002-7699-2064